# Macroevolutionary dynamics of gene family gain and loss along multicellular eukaryotic lineages

Mirjana Domazet-Lošo [1] ✉, Tin Široki[1], Korina Šimičević[1] & Tomislav Domazet-Lošo [2,3] ✉

The gain and loss of genes fluctuate over evolutionary time in major eukaryotic clades. However, the full profile of these macroevolutionary trajectories is still missing. To give a more inclusive view on the changes in genome complexity across the tree of life, here we recovered the evolutionary dynamics of gene family gain and loss ranging from the ancestor of cellular organisms to 352 eukaryotic species. We show that in all considered lineages the gene family content follows a common evolutionary pattern, where the number of gene families reaches the highest value at a major evolutionary and ecological transition, and then gradually decreases towards extant organisms. This supports theoretical predictions and suggests that the genome complexity is often decoupled from commonly perceived organismal complexity. We conclude that simplification by gene family loss is a dominant force in Phanerozoic genomes of various lineages, probably underpinned by intense ecological specializations and functional outsourcing.

New genes are continuously added to genomes through evolutionary time[1–4]. The mechanisms of their formation and their adaptive significance are widely discussed[1–6]. However, the lifecycle of genes also includes gene loss[7], which is comparably much less studied process[8–10]. The recent work reveals that the reductive evolution by gene loss, in parallel with gene gain, played a major role in the evolution of animals[11–15]. However, these studies do not cover phylogenetic nodes deeper than the unicellular ancestors of animals[11,13,14] or, in some instances, the last common ancestor of fungi and animals[12,15]. The deepest node reached so far is the split between eukaryotic lineages Amorphea and Diaphoretickes[16], but this early work is not directly comparable to later studies[11–15], because it used a non-standard sequence similarity search algorithm, which is substantially less sensitive compared to BLAST.

The recent comparative studies in fungi and plants found that genomic gain-and-loss patterns also played important roles along diversification of these major eukaryotic lineages[15,17]. However, similarly to the currently available studies in animals, these analyses have limited phylogenetic depth. The ancestor of Opisthokonta (the animal-fungi clade)[15] and Archaeplastida (a eukaryotic clade containing primary plastids)[17] are the deepest nodes considered in fungal and plant lineages respectively. This leaves the gene gain-and-loss patterns in evolutionary earlier nodes—i.e., the origin of cellular organisms, archaeal lineage and eukaryogenesis period—completely unexplored. Moreover, tracing the gene gains on the phylogenies that are not rooted at the last universal common ancestor (LUCA), and that do not cover archaeal diversifications and eukaryogenesis, inherently leads to uncertainty that the mapped genes, and consequently their functional properties, have potentially deeper evolutionary origin.

For instance, it was thought that Hanks-type protein kinases were eukaryotic innovation. However, a careful screen of bacterial and archaeal genomes for these genes and the comparison of their protein structures reveals that this gene family has its origin in LUCA[18]. Similar argumentation applies to gene losses, which could be reliably detected in the evolutionary younger nodes only if the gene content in all evolutionary older nodes is estimated first. Obviously, to get a

[1]Department of Applied Computing, Faculty of Electrical Engineering and Computing, University of Zagreb, Unska 3, HR-10000 Zagreb, Croatia. [2]Laboratory of Evolutionary Genetics, Division of Molecular Biology, Ruđer Bošković Institute, Bijenička cesta 54, HR-10000 Zagreb, Croatia. [3]School of Medicine, Catholic University of Croatia, Ilica 242, HR-10000 Zagreb, Croatia. ✉e-mail: mirjana.domazet-loso@fer.hr; tdomazet@irb.hr

coherent picture on the gene gain-and-loss patterns, it is essential to consider the full evolutionary trajectory of a focal eukaryotic lineage. This is especially important if one aims at exploring the general trends of genome complexity change.

The macroevolutionary patterns of gene gain and loss are inseparable from the evolution of genome complexity—another long-standing question that in principle could be addressed by reconstructing the content of ancestral genomes[8,10]. Previous work that aimed to recover ancestral genome content was largely restricted to specific phylogenetic nodes that represent, for example, eukaryotic[19], holozoan[11,20], and metazoan ancestor[11,14,21]. In these attempts to estimate the ancestral genome complexity, researchers used various genome features like introns[19,20], orthologous genes[11,13–15,17,20,21], and protein domains[12,20].

However, a comprehensive effort that would encompass the full macroevolutionary profiles from the origin of life to extant organisms is still missing. The lack of such information largely precludes testing the proposed models of genome complexity evolution[8,10,22]. According to the biphasic model, the episodes of rampant increase in genome complexity that are achieved by gene gain are followed by protracted periods of genome simplification through gene loss[8]. Similarly, the complexity-by-subtraction model predicts initial rapid increase of complexity followed by decrease toward an optimum level over macroevolutionary time[22]. However, some authors do not agree with these views and suggest that interactions between simplification and complexification are not predictable[10].

The recovering of gene gain-and-loss patterns on deep phylogenies with hundreds of terminal taxa is further limited with the speed and sensitivity of sequence similarity search algorithms. This is likely the primary reason why previous studies did not consider the full phylogenetic trajectory of their focal eukaryotic groups. However, the recent development of next-generation sequence similarity tools[23–25] which, by keeping comparable sensitivity, substantially outperform older solutions like BLAST in terms of speed—opens opportunity for recovering gene gain-and-loss patterns on a larger collection of genomes. Another uncertainty in reconstructing genomic gain-and-loss patters relates to the choice of evolutionary units that are tracked down. For instance, one could opt to follow evolutionary patterns of protein domains, orthologous genes, gene duplicates (paralogs), or gene families (homologs). The choice will largely depend on which evolutionary question one aims to address.

So far, gain-and-loss studies mainly focused on tracing orthogroups[11,13–15,17,21], presumably because it is assumed that orthologous genes are functionally more conserved than other homologs, although this view is repeatedly challenged[26,27]. These studies use comparable bioinformatic pipelines where all-versus-all protein sequence similarity search is filtered for reciprocal best hits, which are then clustered using the MCL algorithm[28]. However, there are some terminological disparities related to this approach because some of these studies designate their protein clusters as homologous groups[13,17,21], while others refer to them as orthogroups[11,14,15]. We think that the term "orthogroups" fits best the nature of clusters obtained by this type of pipelines because all of these studies restrict protein sequence matches to reciprocal best hits.

Nevertheless, we and others previously showed that the origin of novel genes via significant shifts in the protein sequence space—which could be recovered by unrestrictedly tracing homologs (gene families), thus avoiding reciprocal best hits filtering—also carry the footprints of evolutionary important adaptive events[29–36]. In turn, the underlying macroevolutionary information could be extracted by various statistical approaches[31,32] including functional enrichment analysis[29,33,36].

With an aim of getting a broader perspective on gene family gain-and-loss patterns and to test genome complexity dynamics across the tree of life, here we analyzed evolutionary lineages that start at the common ancestor of cellular organisms and end up in 352 eukaryotic species. To simplify presentation, we performed in-depth functional analyses in four lineages that represent the diversity of deuterostomic animals (*Homo sapiens*), protostomic animals (*Drosophila melanogaster*), plants (*Arabidopsis thaliana*), and fungi (*Saccharomyces cerevisiae*).

In this work, we demonstrate that gene family content in major eukaryotic lineages peaks at major evolutionary and ecological transitions before gradually declining towards the present. This finding supports current theoretical predictions on macroevolutionary change and indicates that genome complexity decouples from organismal complexity in a predictable way. To account for these phenomena, we introduce the idea of "functional outsourcing". This concept implies that any function which is costly for an organism to perform, and at the same time can be outsourced through biological interactions, is a potential target for reductive evolution. We conclude that the dominant force in Phanerozoic genomes across major eukaryotic lineages is the loss of gene families, likely driven by intense ecological specializations and functional outsourcing.

## Results

### Gene family content along eukaryotic lineages

To cover all nodes on the consensus phylogeny that embraces 352 eukaryotic species (Supplementary Data 1), we retrieved 667 reference genomes (see Methods). This number of reference genomes is approximately twofold higher compared to previous studies[11–15,17]. After a multistep contamination cleanup (see Methods), we clustered the amino acid sequences of these reference genomes using the MMseqs2 *cluster* program, a clustering tool which offers a fast and sensitive solution for large protein datasets[23,24]. In contrast to previous studies that considered the gain-and loss patterns of orthologous groups[11,13–15,17] or protein domains on phylogenies[12], we here explicitly traced homologous groups (gene families) by performing sequence similarity searches without filtering for reciprocal best hits[1,26]. We took this perspective since we were interested in recovering significant macroevolutionary changes in the protein sequence space[1,29–31,33,36]; i.e., we aimed to estimate the overall dynamics of gene family diversity on the tree of life.

In this type of analysis, ancient and ubiquitous domains tend to attract multidomain proteins in large clusters, which are then placed at deep phylogenetic nodes[30]. To account for this effect, we varied the $c$ value of MMseqs2 *cluster* algorithm, which sets the minimal fraction of alignment overlap between the reference protein sequence and other cluster members, in the range between 0 and 0.8 (see Methods). On one extreme, a $c$ value of 0.8 forces clustering with an alignment overlap with a cluster's representative sequence of at least 80% protein sequence length[23,24]. In turn, the obtained clusters contain protein sequences with highly similar overall domain architecture. In contrast, a $c$ value of 0 allows clustering without restrictions on the alignment overlap length that consequently leads to the formation of larger clusters, whose members can share only relatively short homologous regions. Expectedly, we find that the total number of gene families in all focal species depends on $c$ values. When we set no restriction on the length of alignment overlap ($c$ value 0), we recovered the lowest number of gene families in total (Supplementary Data 2). In contrast, when we set the alignment length overlap at 80% ($c$ value 0.8), we recovered the highest number of gene families (Supplementary Data 2). As the choice of $c$ value cutoff substantially influences the number of clusters, we performed all our downstream analyses in the full $c$ value range.

Depending on the taxonomic composition of proteins in a cluster, we determined the evolutionary emergence of that gene family and its eventual loss on the consensus phylogeny (Supplementary Data 1, 2). This procedure allowed us to determine a gain-and-loss pattern for every protein family (cluster) recovered in the analyses (see Methods). We then used this information to reconstruct the protein family content of ancestral genomes along the 352 eukaryotic lineages

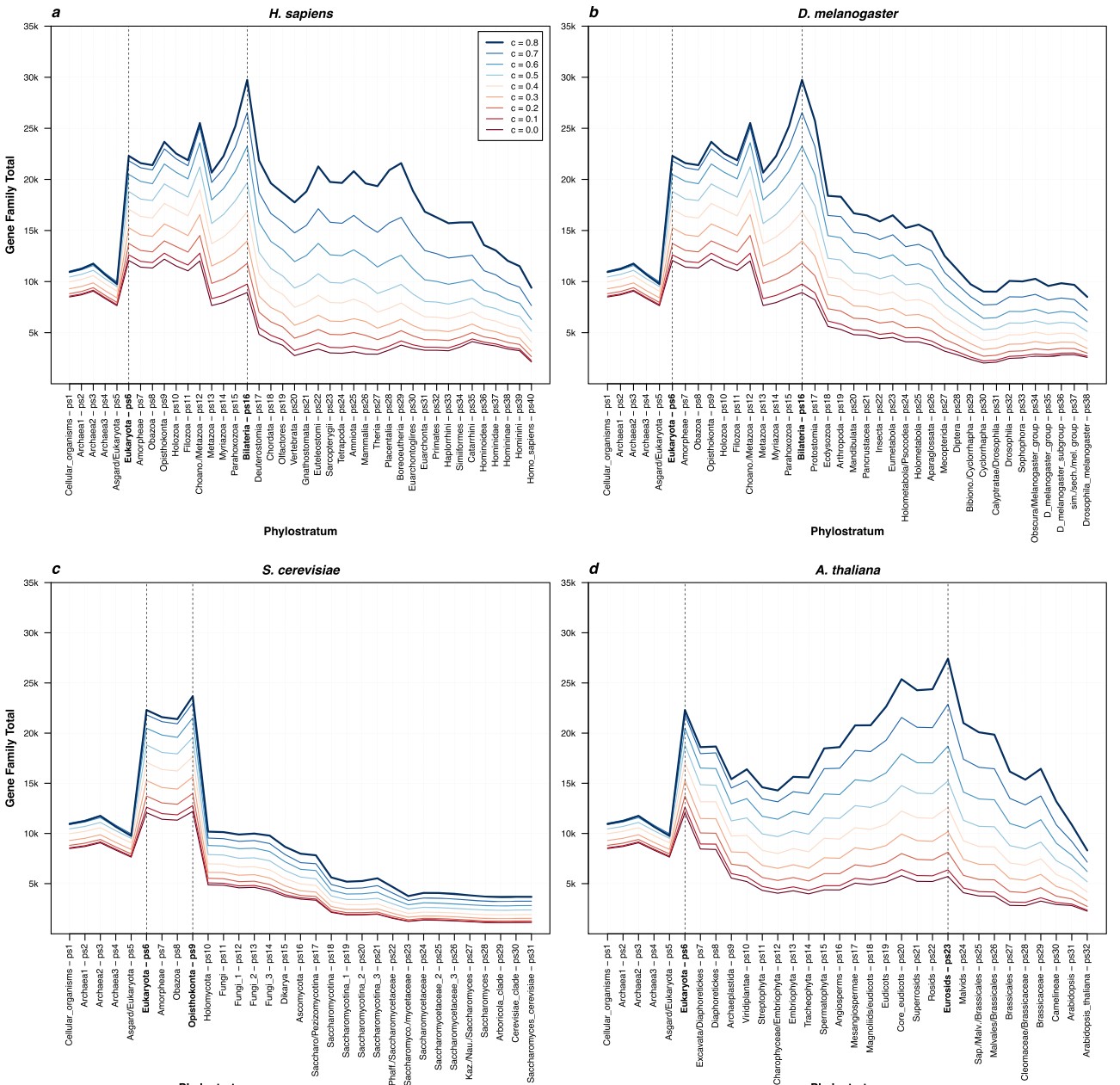

**Fig. 1 | The total number of gene families along the four focal lineages.** The total number of estimated gene families across phylostrata (ps) is depicted for each of the four focal species (**a** *H. sapiens*, **b** *D. melanogaster*, **c** *S. cerevisiae*, **d** *A. thaliana*). The gene family profiles for all 352 eukaryotic species were depicted in Supplementary Data 3. The first phylostratum (ps1) represents the ancestor of cellular organisms and the last one corresponds to a focal species (e.g., ps40 for *H. sapiens*). Colored lines in each plot correspond to different *c* values of the MMseqs2 *cluster* algorithm. This parameter determines the minimal percentage of protein sequence alignment overlap in a cluster. The darkest blue graph corresponds to *c* value = 0.8 which forces at least 80% of sequence length alignment overlap with a cluster's representative sequence. The darkest red graph corresponds to *c* value = 0 which allows clustering without restrictions on the alignment overlap length. The dashed vertical lines on each plot mark the evolutionary range with the highest number of gene families. The source data of this figure are provided in the Source Data file.

(Supplementary Data 3), including the four representative (focal) species (Fig. 1). In all tested eukaryotic lineages, we observed a common evolutionary pattern in which the number of gene families in ancestral genomes sharply increases at the origin of eukaryotes, remains high in early eukaryotic lineages, and then steadily decreases towards extant organisms after reaching a turning point (Fig. 1, Supplementary Data 3). The phylogenetic position of nodes (phylostrata−ps) with the highest number of protein families, depends to some extent on the choice of *c* values. However, regardless of these shifts in the phylogenetic position of the maximal peaks, the overall increase-peak-decrease pattern is always preserved (Fig. 1, Supplementary Data 3).

For instance, in the analyses of *H. sapiens* and *D. melanogaster* lineages (Fig. 1a, b), the maximum number of protein families varies between Eukaryota (*c* value 0, ps6) and Bilateria (*c* value 0.8, ps16). These patterns suggest that the early eukaryotes had the richest catalog of protein families, if one takes the overall homology into account and ignores the precise architecture of multidomain proteins (*c* value 0). In contrast, if the within-gene synteny of homologous regions is tightly controlled by forcing high percentage of alignment overlap (*c* value 0.8), then the first bilaterians show the richest collection of protein families. In either case, the number of protein families more or less continuously decreases after these maximal peaks (Fig. 1a, b). Regardless which criteria for gene family reconstruction we applied,

these patterns suggest that ancestral genomes around the origin of animals and during their early diversification were more complex than the genomes of present animals in terms of gene family diversity. This also holds true if one takes any other animal as a focal species (Supplementary Data 3).

In the analysis of *S. cerevisiae* lineage, we found the maximum number of protein families in the range between Eukaryota (ps6) and Opisthokonta (ps9) (Fig. 1c). The *c* value profiles have very similar shape, which suggests that multi-level reductive processes dominated at the onset and during diversification of the fungal lineage (Holomycota−ps10 to *S. cerevisiae*−ps31, Fig. 1c). This is consistent with patterns in all other fungal lineages (Supplementary Data 3). In contrast, the *A. thaliana* profile is more complex and uncovers that the ancestor of Eukaryota (ps6) had the highest diversity of gene families under permissive clustering (*c* value 0). On the other hand, the stringent clustering, which forces within-gene synteny (*c* value 0.8), reveals that the ancestor of Eurosids (ps23) harbored the highest number of gene families (Fig. 1d). However, all gene family profiles in the *A. thaliana* analysis revealed that the protein diversity loss is especially prominent after the origin of Eurosids (ps23) (Fig. 1d). Highly similar profiles are evident in all other plant lineages as well (Supplementary Data 3).

As an exception, in Brassiceae tribe we found an increased number of gene families in the most recent phylostrata (ps32-ps34, Supplementary Fig. 1), which is a unique feature among plant lineages (Supplementary Data 3). It is well established that the Brassica species underwent multiple autopolyploidization and allopolyploidization events[37,38]. As we strictly trace homologs, our method should not be sensitive to duplication events, including whole genome duplications. The underlying reason is that duplicated genes, no matter how numerous they are, will be simply clustered into parental gene families as long as they retain sequence similarity. However, polyploidization events in Brassiceae tribe obviously led to high diversification rates between paralogs, and consequently to the formation of unusually high number of novel protein sequences; i.e., orphan genes[1,2,6,29,39]. This is an interesting finding that uncovers Brassica species as a hotspot for generation on new gene families.

Our phylostrata (phylogenetic internodes) represent topological levels on the phylogeny, which means that only ordinal evolutionary time, and not the absolute one, can be assessed in this representation type (Fig. 1, Supplementary Data 3). This entails that we can discuss differences in the number of protein families between phylostrata, but we cannot assess how fast these changes happened. Generally, the absolute evolutionary time in our lineages have tendency to be increasingly shorter towards more recent phylostrata (Fig. 1). This raises the question whether the rather stable total number of gene families along the yeast lineage reflects the true long-term stability, or the evolutionary time in this period was simply too short for changes to occur (ps24-ps31, Fig. 1c). The period between the origin of Saccharomycetaceae and the extant yeast (ps24-ps31, Fig. 1c) lasted about 150 My[40]. In comparison, this roughly corresponds to the period between the origin of Eudicots and the present-day *Arabidopsis* (ps19-ps32, Fig. 1d) in the plant lineage[41]. If one assumes comparable evolutionary rates in plant and fungal lineages, this demonstrates that the lack of evolutionary time cannot be an explanation for the Saccharomycetaceae stability in the number of protein families. Actually, the profile of *Brassica* lineage (ps32-ps34, Supplementary Fig. 1) showed that substantial changes in the number of gene families are detectible even in periods shorter than 10 My[39].

When compared to the suggested models of macroevolutionary change in genome complexity, our increase-peak-decrease patterns of gene family diversity fits the best the complexity-by-subtraction model, which envisages initial rapid increase in complexity that leads to a maximum value, followed by complexity decrease over macroevolutionary time towards an optimum[22]. However, our patterns also agree with the biphasic model that predicts the episodes of rampant increase in genome complexity, which are followed by protracted periods of genome simplification[8]. The major difference between these two models is that the biphasic model predicts several waves of genome complexity change over macroevolutionary time[8,22]. In our gene family analysis, we see an obvious two-wave gene family diversity pattern only in plants (Fig. 1, Supplementary Data 3). For instance, in *A. thaliana* lineage the genome complexity measured by gene family diversity first peaks at Eukaryota (ps6, Fig. 1d), decreases towards Streptophyta (ps11, Fig. 1d), peaks again at Eurosids (ps23, Fig. 1d), and then finally rapidly decreases towards focal *A. thaliana* (ps32, Fig. 1d). As an exception, the beginning of a possible third wave is visible in the most recent phylostrata of *Brassica* lineage (ps32-ps34, Supplementary Fig. 1).

It is important to note that the type of recovered evolutionary information, conveyed by gene family profiles in Fig. 1 and Supplementary Data 3, depends on the *c* value. On one extreme, when clustering is performed with *c* value = 0, the acquired gene families reflect the evolution of completely new sequences in the protein sequence space. On the other extreme, when clustering is performed with *c* value = 0.8, the acquired gene families reflect the evolution of specific protein sequence architectures along the most of their sequence length. Accordingly, the highest number of gene families at Eukaryota (ps6, Fig. 1), in profiles generated with *c* value = 0, suggests that evolution through the emergence of entirely new protein sequences were the most pronounced during eukaryogenesis. In contrast, the highest number of gene families, in profiles generated with *c* value = 0.8, at Bilateria (ps16, Fig. 1a, b) and at Eurosids (ps23, Fig. 1d) reveals that the emergence of proteins with strictly defined protein architecture were at peak in these periods of animal and plant evolution respectively (Supplementary Data 3). These results agree with a previous work in animals which detected an increased acquisition of new genes via protein sequence rearrangement processes at the stem of animals[12]. However, in the yeast analysis, there are no apparent shifts in the phylogenetic position of maximal number of protein families when the *c* values profiles of 0 and 0.8 were compared (Fig. 1c). This suggests that in the fungal lineage the emergence of completely new protein sequences and the assembly of proteins with specific sequence architectures were largely in balance (Supplementary Data 3).

We chose our four focal species because they are well-studied model organisms, which allowed us to perform detailed functional enrichment analyses (see below and Methods). However, the choice of yeast as a representative of fungal linage might be problematic, because it is a secondary unicellular organism which experienced substantial losses, and thus might not properly reflect the evolutionary trajectory of fungi as a group. However, all other fungal lineages showed highly similar change in the protein family content of ancestral genomes (Supplementary Data 3). An example is black truffle (*Tuber melanosporum*), a multicellular Pezizomycetes fungi (Supplementary Fig. 2). This fungal clade was also used as a focal lineage in the recent analysis of gain-and-loss patterns in animals and fungi[15]. Accordingly, we conclude that the increase-peak-decrease pattern is a genuine trend in fungi, and not a peculiarity linked to unicellular adaptations of yeasts.

Previous gene gain-and-loss studies typically use MCL-based clustering approaches to assess evolutionary dynamics of orthology-based sequence clusters[11,13–15,17]. In contrast to the MCL-based approaches, the MMseqs2 *cluster* algorithm, which we applied here, offers a two-in-one solution where sequence searches and clustering steps are integrated in an iterative cascade that provides much faster computation[23,24]. This is not a trivial improvement for datasets that include hundreds of reference genomes. Nevertheless, because the MMseqs2 *cluster* algorithm is relatively new and is not commonly used in this type of studies, we tested how the application of an MCL-based approach changes the protein family content of ancestral genomes,

from the perspective of *H. sapiens*. To achieve this, we first made all-against-all comparison using the MMseqs2 *search* algorithm—an independent tool within the MMseqs2 package with BLAST-like behavior[24]—and then in the next step clustered the obtained hits using MCL[28].

Interestingly, the evolutionary profiles of protein family content obtained by the MCL approach are essentially identical to those obtained by the MMseqs2 *cluster* approach (Supplementary Fig. 3). This demonstrates that the ancestral content of protein families, which we presented in Fig. 1 and Supplementary Data 3, are not dependent on the choice of a clustering algorithm. Although the profiles are qualitatively highly similar, the MCL procedure generally returns higher number of clusters (Supplementary Fig. 4). Some previous studies recognized this problem with the MCL algorithm and tried to address it by changing the inflation parameter, which controls this behavior, from the default 2 to 1.5[14]. However, when we applied this more conservative approach, the MCL procedure again returned the higher number of clusters than the MMseqs2 *cluster* algorithm (Supplementary Fig. 3b).

The *c* value approach as implemented in MMSeqs2 inherently controls for possible gene length bias by forcing gene clustering to length bins. However, to further test whether any residual gene-length bias could influence our results, we included a gene length and phylogenetic distance normalization procedure in our MCL pipeline, as described in the OrthoFinder paper[42]. This analysis showed that the OrthoFinder-type gene length normalization generally does not influence the evolutionary trajectory of protein family content (Supplementary Fig. 3c, d). In principle, not accounting for gene length biases could result in an over-splitting of homologous groups. However, our test showed that this effect is negligible, especially for high *c* values (Supplementary Fig. 4). Moreover, the MMseqs2 clustering returned less clusters compared to the MCL clustering whether or not the gene length normalization was included (Supplementary Fig. 4).

A recent work estimates that the last eukaryotic common ancestor (LECA) possessed 10,233 Pfam domains and 12,753 genes[43]. The number of protein families were not directly estimated in this study, but their number should be somewhere between these two values. To evaluate our results against this crude approximation, we considered both extreme c values (0 and 0.8). In our study, LECA corresponds to phylostratum Eukaryota (ps6) in depicted phylogenies (Fig. 1). Our MMseqs *cluster* analysis recovered between 12,065 (*c* value 0) and 22,301 (*c* value 0.8) protein families in Eukaryota (ps6), while the most conservative version of the MCL-based procedure (*I* = 1.5 plus normalization) returned between 21,138 (*c* value 0) and 28,438 (*c* value 0.8) protein families (Supplementary Fig. 4). Obviously, the MMseqs2 *cluster* algorithm recovered much closer values to the Pfam domain-based estimates than MCL-based procedure. Similarly, another study estimated that 11,093 protein families contained homologs from bacteria and archaea[44], which corresponds to the phylostratum cellular organisms (ps1) in our phylogeny (Fig. 1, Supplementary Data 1). The MMseqs2 *cluster* analysis recovered between 8496 (*c* value 0) and 10,942 (*c* value 0.8) protein families. In contrast, the most conservative MCL-based procedure (*I* = 1.5 plus normalization) returned between 13,826 (*c* value 0) and 14,424 (*c* value 0.8) protein families (Supplementary Fig. 4). This shows that the MMseqs2 *cluster* algorithm is more conservative solution than the MCL-based procedure, which have tendency to inflate the number of clusters. Hence, we continued our downstream analysis using clusters produced by the MMseqs2 *cluster* approach.

## Gain and loss ratios

Although our gene family content profiles in Fig. 1, Supplementary Data 3 and Supplementary Fig. 4 are the result of a careful bioinformatic analysis in a broad parameter space, the clustering of evolutionary distant reference genomes in large numbers and with new algorithms could possibly lead to unforeseen biases that obscure evolutionary signals and distort biological reality. We therefore made a more detailed evaluation of our gene family datasets, compared recovered patterns to findings in previous studies, and looked for expected and novel biological imprints. First, we explored the ratios of gene family gain and loss across lineages that lead to our four focal species (Fig. 2, Supplementary Data 4). We found substantial loss events at the origin of metazoans (ps13) and deuterostomes (ps17) in the analysis that takes *H. sapiens* as a focal species (Fig. 2a). Similarly, on the lineage leading to *D. melanogaster*, we found extensive loss events at the origin of metazoans (ps13), protostomes (ps17), and ecdysozoans (ps18) (Fig. 2b). These results fully agree with previous studies which, by tracing orthologs groups, detected major loss events at the origin of metazoans[11,13], deuterostomes[13,14], protostomes[13,14], and ecdysozoans[13,14].

The gain-and-loss profile in the *S. cereviseae* lineage reveals a dominant loss event at Holomycota (ps10) and a largely reductive trend in subsequent phylostrata (Fig. 2c). This pattern agrees with previous work which, by tracing orthologs, detected substantial reductive genome evolution in the pre-fungal ancestors (Holomycota)[15] and along the fungal lineage[15,40,45]. On the other hand, our analysis of the *A. thaliana* lineage showed an apparent two wave pattern (Figs. 1, 2d), where the first wave of dominant gene family losses stretches between Excavata/Diaphoretickes and Streptophyta (Fig. 2d, ps7–ps11), and the second one between Malvids and Arabidopsis (Fig. 2d, ps24–ps32). Interestingly, along the diversification of land plants (Embryophyta to Eurosids, ps13–ps23) we detected substantial increase in gene family gains (Figs. 1d, 2d). This pattern is in accord with the most recent gain-and-loss analysis in plants which detected an increasing number of novel genes at the origin of land plants[17].

Taken together, these comparisons reveal that our gene family gain-and-loss ratios (Fig. 2, Supplementary Data 4) generally corroborate findings gained by previous studies. We take this as an additional argument that our complexity profiles of protein family content in ancestral genomes reflect biological reality (Fig. 1, Supplementary Data 2, Supplementary Data 3). Compared to our approach, previous studies use different pipelines in terms of sequence similarity search algorithms, homology restrictions and clustering strategy. This suggests that the major macroevolutionary signals, which could be recovered by phylogeny-aware clustering approaches, are highly resilient to the choice of the methodological pipelines. Interestingly, the ratios of gene family gain and loss are generally very rarely in balance along four focal lineages (Fig. 2, Supplementary Data 4). This produces erratic pattern and in turn suggests that organisms alternate between the predominantly expanding and reductive mode of protein family evolution. However, an antagonistic mechanism that connects gene family gain and loss processes is as yet unknown, which might be an interesting topic for future studies.

## COG functional enrichments

To this point, we have quantified changes in the numbers of gene families across macroevolutionary time. However, a probably more interesting question is how gene family functions overlay these quantitative profiles. Thus, we next sought to map functional data on our clusters and to explore if the obtained functional patterns are evolutionary meaningful. We first mapped COG functional categories to our phylogenetic patterns of gene family gain and loss, and then we explored the enrichment profiles of every COG functional category across phylostrata (Fig. 3, Supplementary Data 5).

For instance, we detected in the *H. sapiens* analysis five groups of COG functions that show phylogenetically distinct enrichment profiles in gained gene families. The first group are metabolic functions (N, H, F, M, C, E, Q, G, V, P, I) that showed strong enrichment signal at the origin of cellular organisms (ps1, Fig. 3). The second group are

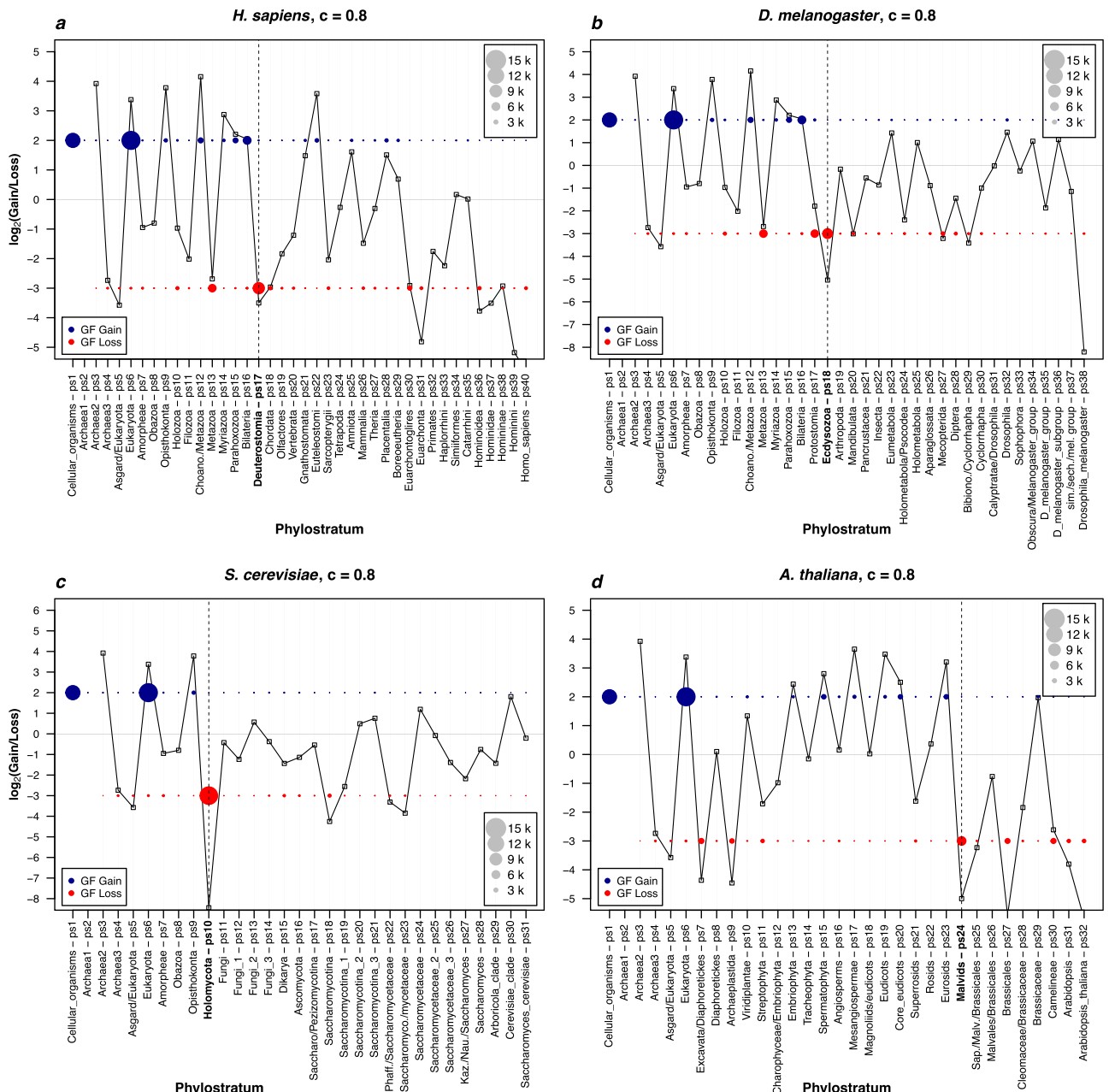

**Fig. 2 | Evolutionary dynamics of gene family gain and loss ratios.** The $\log_2$ values of the gene family gain and loss ratio across phylostrata for each of the four focal species (**a** *H. sapiens*, **b** *D. melanogaster*, **c** *S. cerevisiae*, **d** *A. thaliana*) are shown. We recovered gene families using the *c* value cutoff of 0.8. Figures with all *c* values cut-offs (0–0.8) are shown in Supplementary Data 4. The first phylostratum (ps1) represents the ancestor of cellular organisms and the last one corresponds to a focal species (e.g., ps40 for *H. sapiens*). The vertical dotted lines mark the phylostrata with the major gene family loss events that precede the changing trends in the total number of protein families (Fig. 1, *c* value 0.8). The circles of different sizes represent the number of gained (blue) and lost (red) gene families in a respective phylostratum. Letter k in the legends stands for kilo. The third phylostratum from the root is the first phylostratum where the loss events in a particular focal lineage can be calculated (see Methods). The source data of this figure are provided in the Source Data file.

information processing functions (J, K, L) that showed enrichments along the divergence of Archaea (ps1-ps5, Fig. 3). The third group are functions specific for a eukaryotic cell (A, Y, B, D, U, O) that showed enrichment at the origin of Eukaryota (ps6, Fig. 3). The fourth group of functions (T, signal transduction; W, extracellular structures), which are particularly related to animal multicellularity, have protracted enrichment profile that starts in pre-metazoan ancestors (ps11, ps12, Fig. 3) and continues along animal radiation up to the origin of mammals (ps26, Fig. 3). Finally, clusters without annotations show expected pattern with high enrichment in younger phylostrata (ps21–ps38, Fig. 3)[2,46]. The three of these functional enrichment periods—the origin

of the cellular organisms, the origin of eukaryotes and the origin and diversification of metazoans—essentially mark major evolutionary transitions in the evolutionary trajectory leading to extant animals[47].

On the other hand, it is interesting to note that enriched functional gains generally precede enriched functional losses of the same function (Fig. 3, Supplementary Data 5). For instance, most of the metabolic functions (N, H, F, M, C, E Q, G) showed a very strong enrichment signal at the origin of cellular organisms (ps1, Fig. 3), which is followed by protracted loss along archaeal ancestors (ps3-5, Fig. 3), at the origin of Eukaryotes (ps6-7, Fig. 3), and in pre-metazoan and early metazoan lineages (ps10-17, Fig. 3). The similar pattern, where

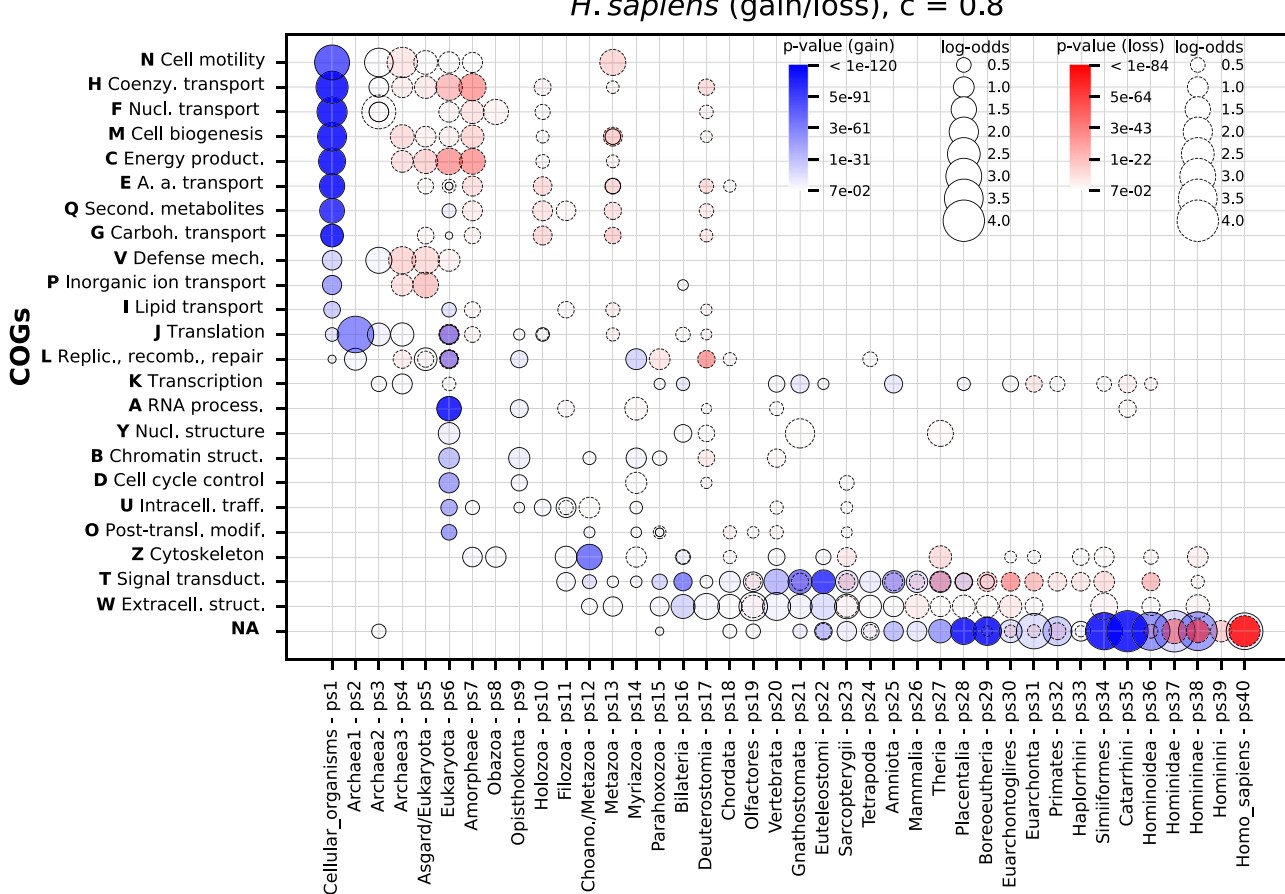

**Fig. 3 | The enrichment of COG functional categories in gained and lost gene families along the *H. sapiens* lineage.** The abbreviated names of COG functional categories and corresponding one-letter symbols are depicted at y-axis. The protein families without COG annotation are annotated with NA. The names and symbols of phylostrata are shown on the x-axis. The gene families are reconstructed with the MMseqs2 *cluster* program using a *c* value of 0.8. Figures with *c* values in the range between 0 and 0.8 for four focal species are in Supplementary Data 5. Functional categories significantly enriched among gained gene families across phylostrata are depicted by solid circles painted in shades of blue that reflect *p* values. Functional categories significantly enriched among lost gene families across phylostrata are depicted by dashed circles painted in shades of red that reflect *p* values. The size of solid and dashed circles is proportional to enrichment values estimated by log-odds. The significance of enrichments was estimated by two-tailed hypergeometric test corrected for multiple testing. The third phylostratum from the root is the first phylostratum where the loss events in a particular focal lineage can be calculated (see Methods). The source data of this figure are provided in the Source Data file.

functional gains are followed by losses, is also evident in functions specific for eukaryotes (A, Y, B, D, U, O) and metazoans (T, W). In addition, the statistical strength of enrichment signals (*p* value) is generally higher in gained than in lost enrichments per function (Fig. 3).

These trends suggest that the major evolutionary transitions are underpinned by massive functional gains, which are than streamlined by partial losses during the radiation of respective lineages. Many functions in the other three focal species also showed this type of bias in comparison between loss and gain functional enrichments (Supplementary Data 5). These regularities could also be viewed through the lens of the complexity-by-subtraction and biphasic models of macroevolutionary change[8,22]. From this perspective, an initially strong increase in functional complexity is often followed by a stepwise process of functional simplification. It seems that, in addition to the sequence complexity level (Fig. 1), similar principles operate at the level of functional complexity (Fig. 3).

The enrichment analysis of COG functional categories allowed us to get better insights into some prominent events detected in the gain-and-loss ratio analysis (Fig. 2). For instance, gene families lost at the origin of deuterostomes (ps17, Fig. 2a) are enriched with a wide range

of COG categories (13 categories, *c* value 0.8), with prevalence of metabolic functions (ps17, Fig. 3). This contrasts the gene family loss at the origin of protostomes (ps16, Fig. 2b), where lower number of COG categories is enriched (7 categories, *c* value 0.8), with prevalence of functions related to the performance of eukaryotic cell and animal multicellularity (Supplementary Fig. 5). This reveals that the massive losses of gene families, which were present in both deuterostomes and protostomes after the divergence from the last bilaterian ancestor (Fig. 2), are functionally distinct. This in turn suggests that the ancestors of these two major bilaterian groups might have occupied profoundly different ecological niches. Another interesting finding is a strong enrichment signal of cytoskeleton-related functions (COG category Z) at the origin of Choanozoa (ps12, Fig. 3), which we analyzed in more details below. In brief, these global COG profiles reassured us that our clustering approach is evolutionary and biologically meaningful, hence we further refined the functional analysis by mapping fine-grained Gene Ontology (GO) terms on our clusters.

**GO functional enrichments**
The full overview of profiles with enriched GO functions along our four lineages is depicted in Supplementary Data 6–9 (gain) and

Supplementary Data 10–13 (loss). Using the same protocol as in the COG functional enrichment analysis, we tested if some evolutionary period (phylostratum) has higher frequency of a specific GO functional annotation than would be expected by chance alone for a particular focal lineage. This type of enrichment analysis detects evolutionary periods that were adaptively important for a specific functional category from the perspective of a tested focal lineage. Interestingly, our analysis yielded approximately between 5000 and 11,000 statistically significant GO enrichment profiles per focal species (Supplementary Data 6–13). This large collection of enriched GO functions shows that functional information is not randomly distributed over our clusters, and that our clustering approach generally provides a suitable platform for recovering functional evolutionary patterns.

In the further evaluation of these results, we first noticed that there are around two to three times more GO functional categories that are significantly enriched among gained compared to lost gene families (Supplementary Table 1, Supplementary Data 14). However, because not all gene families have gene loss events–i.e., some gene families are retained up to a focal species–the number of functional enrichments in the gene family gain and loss pools are not directly comparable. To adjust for this, we excluded from the analysis gene families that have only gain events (Supplementary Table 1, Supplementary Data 14). In this way we were able to contrast the number of functional enrichments between gain and loss events using the same gene family pool. Regardless of these corrections, we again found the higher number of significantly enriched GO functions among gain events (Supplementary Table 1, Supplementary Data 14).

This robust asymmetry indicates that gene family gains are more often functionally non-randomly distributed over phylogeny compared to gene family losses. In other words, gene family gain events tend to functionally group together on the phylogeny. This contrasts gene family loss events, whose functions are more often randomly distributed, and thus are not detected in the enrichment analysis. In the context of discussion on the genome complexity, this is an important finding because it supports the prediction of the biphasic model that genome simplification by gene loss is largely neutral process that occurs roughly in a clock-like manner[8], in contrast to genome complexification by gene gain which usually occurs abruptly and is associated with evolutionary adaptations[8,29].

After considering these global functional enrichment patterns, we sorted out some prominent functional categories, which demonstrate that our clustering approach and phylostratigraphic mapping recovers relevant biological information. First, we dissected the most striking increase in the number of gene families at the origin of eukaryotes (ps6 in all lineages, Figs. 1, 2, Supplementary Data 3). We recovered significant enrichment signals at the origin of eukaryotes, related to gene family gains, for essentially all functions that are proposed in the literature to be eukaryotic defining properties[10,48,49] (Fig. 4, Supplementary Fig. 6, Supplementary Data 15). These include nucleus, endomembrane system, cytoskeleton and motility, endosymbiont, sexual reproduction and other eukaryotic specific features (Fig. 4, Supplementary Fig. 6, Supplementary Data 15). This result indicates that the abrupt increase in the number of protein families that we detected at Eukaryota (ps6, Figs. 1, 2) reflects the burst of innovations related to cell organization that occurred during eukaryogenesis[10,50,51].

Besides these expected patterns for eukaryogenesis-related functions, we also uncovered some additional signals that are intuitive, but previously undetected. As we already explained, we found a strong enrichment signal of cytoskeleton-related functions (the COG term Z) at the origin of Choanozoa (ps12, Fig. 3). In contrast to the COG functional annotations, which are based on general terms and limited vocabulary, GO annotations allowed us a more specific analysis. First, we tested whether the two annotations systems (COG and GO) return congruent results by looking at the enrichment pattern for the GO term cytoskeleton (GO:0005856), which has the closest meaning to

the COG term cytoskeleton (Z). Indeed, we found a strong enrichment signal at the origin of Choanozoa (ps12), which is detectable from the perspective of *D. melanogaster* and *H. sapiens* as focal species (Supplementary Fig. 7).

We then explored more specific cytoskeleton-related GO terms in *D. melanogaster* and *H. sapiens* dataset and found strong enrichment signals at the origin of Choanozoa (ps12) for cilium organization (GO:0044782) (Fig. 5), cilium movement (GO:0003341), microtubule cytoskeleton (GO:0015630) and axoneme (GO:0005930) (Supplementary Figs. 8, 9, 10). The comparison of enrichment patterns for these GO terms between the four lineages is a good example that our protocol for calculating functional enrichments detects lineage-specific adaptations (Fig. 5, Supplementary Figs. 8, 9, 10). Namely, although the four focal species share the phylogeny up to the origin of Eukaryotes (ps6, Fig. 5), and consequently the content of protein families for that part of the phylogeny is the same, the statistical recovery of the enrichments in this period will vary because the deviations from the expected frequencies are calculated in reference to the entire span of a focal lineage. For instance, from the perspective of *D. melanogaster* and *H. sapiens* as focal species, we detected strong enrichment signals at Choanozoa (ps12, Fig. 5a, b), which partially overshadow those at the origin of Eukaryota (ps6, Fig. 5a, b). In contrast, the enrichment signals at the origin of Eukaryota are more prominent from the yeast and *A. thaliana* perspective because no major adaptive gain events related to flagellar/ciliary structures occurred in these lineages after the origin of eukaryotes (ps6, Fig. 5c, d). These patterns together suggest that the flagellar/ciliary apparatus of choanozoans, compared to other eukaryotes, is a derived feature underpinned by the recruitment of novel protein families[52].

This is suggestive finding because the collar complex, an apical flagellum surrounded by a funnel-like collar of microvilli, is a cytoskeleton-based synapomorphy of choanozoans[53]. Moreover, the microvillar collar, as the most prominent part of this structure, was used as a basis for the taxonomical naming of this clade. This important cell-level adaptation facilitates cell movements and allows an efficient collection of bacterial pray in choanoflagellates[53]. In metazoans, this adaptation is retained in choanocytes and, in a derived form, in various other cell types that possess microvilli[54,55]. Unfortunately, the GO collection does not contain specific GO terms such as "collar complex" or " funnel-like collar", which would allow us to directly trace the functional evolution of this outstanding trait as a whole. However, we recovered the enrichment profiles of an intimately related subordinate term stereocilium (GO:0032420); a specialized form of microvilli present in the sensory cells of some metazoan groups[55,56]. This functional annotation showed a unique signal at the origin of Choanozoa (ps12, Supplementary Fig. 11), which gives further support to the notion that microvilli[54,55], and consequently collar complex[53], are crucial innovations of choanozoans.

## Lineage specific GO functional enrichments

To further confirm the biological relevance of our gene family reconstruction, we explored GO functional enrichments that are specific for each of our four focal lineages. For instance, the emergence of adaptive immunity in vertebrates left a strong imprint in the evolutionary period that represents early vertebrate radiation (Fig. 6, Vertebrata to Placentalia, ps20-ps28), with especially prominent signals at the origin of jawed vertebrates (Fig. 6, Gnathostomata, ps21). These functional enrichment patterns are in line with the view that the origin of adaptive immune system in vertebrates is a major evolutionary innovation[57,58]. Similarly, metamorphosis is considered a key innovation that contributed to biological success of insects[59]. The enrichment profiles in *D. melanogaster* lineage revealed that the insect metamorphosis has deep roots in animal phylogeny and that it was continuously reshaped by the recruitment of novel protein families along the insect radiation (Fig. 7, Bilateria to Cyclorrhapha, ps16-ps30). Interestingly, we also

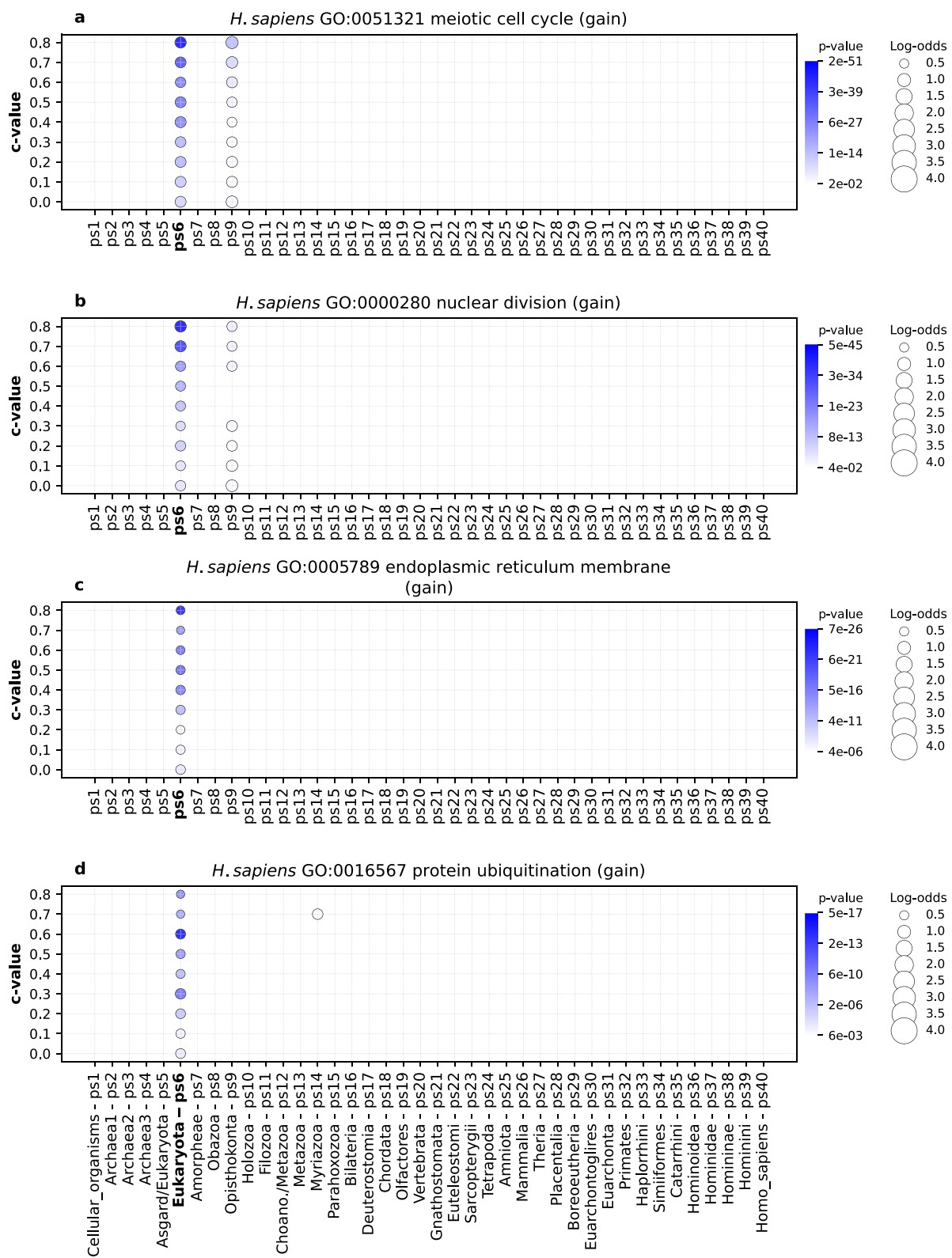

detected significant loss of protein families related to metamorphosis in *D. melanogaster* (Fig. 7, ps38) in line with the previous findings of substantial gene family loss in this species[60].

A striking example of reductive evolution is the loss of respiratory chain complex I in the oxidative phosphorylation pathway of Saccharomycetaceae yeasts related to their fermentative and anaerobic lifestyles[61,62]. This loss was initially detected by orthology tracking[61];

however, our protein family-based approach also recovered a statistically significant loss of functions associated to respiratory chain complex I at Saccharomycetaceae *sensu lato* (ps23, Fig. 8, Supplementary Data 1).

Similarly to the other three focal species, we recovered the enrichment signals of many functions that mark the biology of the *A. thaliana* lineage (Supplementary Data 1, 9, 13). Plastids—which

**Fig. 4 | An example of GO functional enrichments related to eukaryogenesis in the *H. sapiens* lineage.** The results are shown for four GO terms: **a** GO:0051321 (meiotic cell cycle) **b** GO:0000280 (nuclear division) **c** GO:0005789 (endoplasmic reticulum membrane) **d** GO:0016567 (protein ubiquitination). Functional enrichments were calculated using the set of gained gene families along the *H. sapiens* lineage (x-axis). The extended catalog of eukaryogenesis-related terms is listed in Supplementary Fig. 6 and their corresponding enrichment charts are shown in Supplementary Data 15. The gene families are reconstructed with MMseqs2 *cluster* using a range of *c* values (0 to 0.8, y-axis). Solid circles depict significant enrichments of a GO term in gene families gained at a particular phylostratum. The size of circles is proportional to an enrichment value estimated by log-odds, while the shades of blue correspond to *p* values. The significance of enrichments was estimated by two-tailed hypergeometric test corrected for multiple comparisons. Only enrichments with $p < 0.05$ are shown. The four depicted GO terms showed a consistent enrichment at Eukaryota (ps6) across tested *c* values. The source data of this figure are provided in the Source Data file.

originated by an endosymbiotic event involving cyanobacteria and eukaryotic host (Archaeplastida)—are arguably the most important innovation that allowed radiation and ecological expansion of the plant lineage[63]. Our enrichment analysis uncovers continuous gain of protein families with plastid related functions in the broad period from the origin of Diaphoretickes to Eurosids (ps8-ps23, Fig. 9). The first signals from Diaphoretickes to Viridiplanteae (ps8-ps10, Fig. 9) reflect the phylogenetic distribution of plastids that were independently acquired by different lineages that branch within Diaphoretickes[64,65]. In these situations where a feature (e.g., plastids) was acquired more than once through horizontal transfer, our methodology, which is based on Dollo's parsimony, is not able to establish the exact origin of that particular feature.

Nevertheless, our approach correctly bracketed phylogenetic presence of plasmids (ps8-ps10, Fig. 9), and uncovered strong enrichment signals in the later periods where plastids are inherited exclusively vertically. These later enrichments suggest a protracted and intensive coevolution of plastids and host cells at the origin of land plants (Streptophyta-Embriophyta, ps11-ps13, Fig. 9) and along its diversification (Spermatophyta-Eurosids, ps15-ps23, Fig. 9). A prominent example of plastid evolution within angiosperms is the emergence of chromoplasts, a derived plastid type, which confers bright colors to flowers and fruits[63]. In accordance with this, our analysis uncovered strong enrichment of gene families related to chromoplasts during early diversification of flowering plants (ps18, Fig. 9) and especially at core Eudicots (ps20, Fig. 9).

Finally, these lineage-specific functional profiles we singled out are only a tiny portion of those available in Supplementary Data 6–13, which could serve as a valuable resource for researchers interested in functional macroevolutionary patterns.

## Discussion
Taken together, the functional signals that we recovered in our gain-and-loss phylostratigraphic maps demonstrate biological validity of our gene family reconstruction, which shows the increase-peak-decrease pattern of protein family diversity at the macroevolutionary scale (Fig. 1, Supplementary Data 3). The shape of these protein diversity trajectories largely agrees with the predictions of biphasic and complexity-by-subtraction models on genome complexity evolution[8,22]. However, it remains unclear which evolutionary forces produce these large-scale increase-peak-decrease protein diversity patterns that appear in independently evolving eukaryotic lineages. There is an extensive discussion on this topic in previous work that tries to weigh the relative importance of adaptive and neutral processes[8,10,22]. The higher number of functional enrichments that we found in gene family gain compared to gene family loss events (Supplementary Table 1) suggests that gene family losses are more frequently result of neutral processes. Yet, numerous functional enrichments that we found in gene family loss patterns (Supplementary Data 10–13) indicate that some reductive events are also the result of adaptive evolution. For example, the loss of respiratory chain complex I in yeasts (Fig. 8) is likely an adaptive event, although at present it can be only speculated which selective pressures drove this reduction[10,61,62].

Nevertheless, on the gross scale it is very indicative that the reductive trend in our lineages correlate with important ecological transitions. For instance, after the split of deuterostomic and protostomic animals the complexity of their genomes at the level of gene families have been continuously reduced until the present time (Fig. 1a, b, Supplementary Data 3). This switch to the reductive mode of genome evolution corresponds to Proterozoic-Cambrian transition, where profound abiotic and biotic environmental changes occurred, which, in turn, allowed animal radiation and the complexification of marine ecosystems[66–68]. One could speculate that the rise in complexity of feeding ecology in this geological period, which included the evolution of more efficient predation modes, together with the emergence of new and more specialized ecological niches[66–68], created possibilities for the reduction in gene family complexity. A very illustrative example in this regard is the loss of capability to synthetize nine essential amino acids at the root of animal tree[11]. This massive metabolic simplification, which is a synapomorphy of animals, is most likely linked to the diet changes that allowed the animal ancestor to efficiently get these amino acids from the environment. More efficient filter feeding and/or predation are innovations that probably led to this reduction[69].

A further conceptual support for this idea comes from the free-living communities of planktonic bacteria where some members provide costly and indispensable functions as public goods. Other bacterial species in the community exploit the environmental availability of these functions and benefit by simplifying their genomes via the loss of costly genetic machineries[70]. The Black Queen Hypothesis (BQH), which models this phenomenon, predicts that the loss of such leaky and costly functions is selectively favored, and that it leads to the emergence of new and long-lasting biological dependencies between bacterial species[70]. This idea could be generalized in a way that any function that is costly to perform for an organism, and at the same time could be outsourced through biological interactions, is a potential target of reductive evolution. These processes could then leave an imprint in the genome by decreasing gene family diversity of an organism. We term this generalization "functional outsourcing". For instance, changes in feeding ecology, niche specialization, and increasingly more intimate interactions within animal holobionts[71], could all have triggered the reduction in protein family diversity within particular animal lineages.

Interestingly, if this idea holds true, our recovered patterns of continuous reduction in protein family diversity during Phanerozoic period (Fig. 1a, b, Supplementary Data 3) signal that the complexity and strength of biological interactions, which involve animals, were more or less continuously increasing in the last 540 million years. In other words, gene family diversity (complexity) is negatively correlated with the strength (complexity) of biological interactions. This sheds some light on the problem of measuring complexity of biological systems. It is well known that organismal complexity, at various phenotypic levels, frequently does not reflect genomic complexity, and vice versa[8]. Wolf and Koonin proposed that genomic complexity, measured by the number of conserved genes, could be complemented with other measures to obtain a better proxy of organismal complexity[8]. To our knowledge there have not been any further attempts in this direction so far.

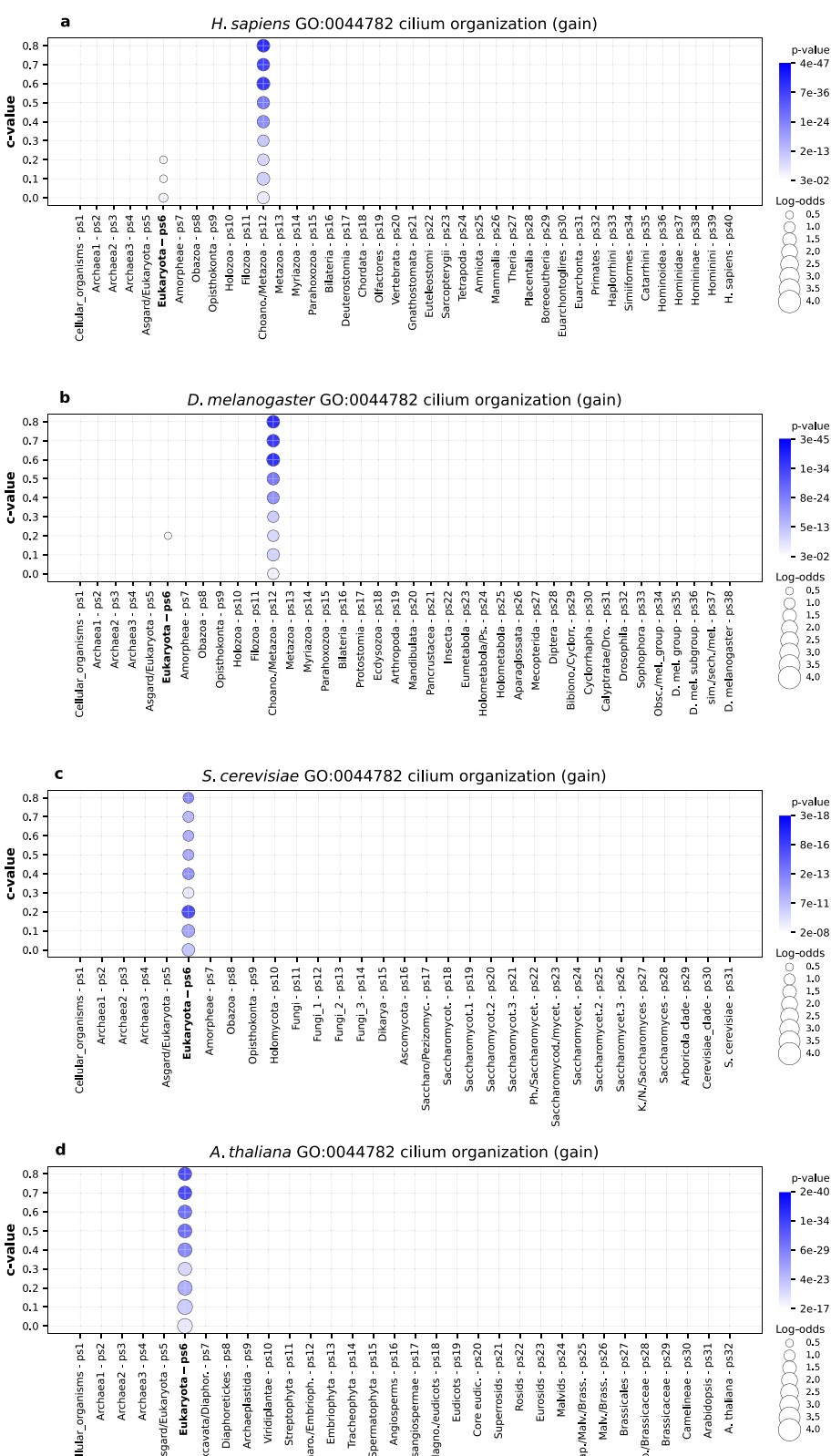

**Fig. 5 | The enrichment of GO functional categories related to cilium organization in the four focal species.** The enrichment profiles are shown for the GO term GO:0044782 (cilium organization). The functional enrichments were calculated using the sets of gained gene families along **a** *H. sapiens*, **b** *D. melanogaster* **c** *S. cerevisiae* and **d** *A. thaliana* lineages (x-axis). The gene families are reconstructed with MMseqs2 *cluster* using a range of *c* values (0–0.8, y-axis). Solid circles depict significant enrichments of the GO term in gained gene families at a particular phylostratum. The size of circles is proportional to enrichment values estimated by log-odds, while the shades of blue (gain) correspond to *p* values. The significance of enrichments was estimated by two-tailed hypergeometric test corrected for multiple comparisons. Only enrichments with *p* < 0.05 are shown. This GO term shows the significant enrichments signals at the origin of Eukaryota (ps6) in the four focal species (**a**–**d**), with prominent additional signals at Choanozoa (ps12) in the *H. sapiens* (**a**) and *D. melanogaster* (**b**) analysis. The source data of this figure are provided in the Source Data file.

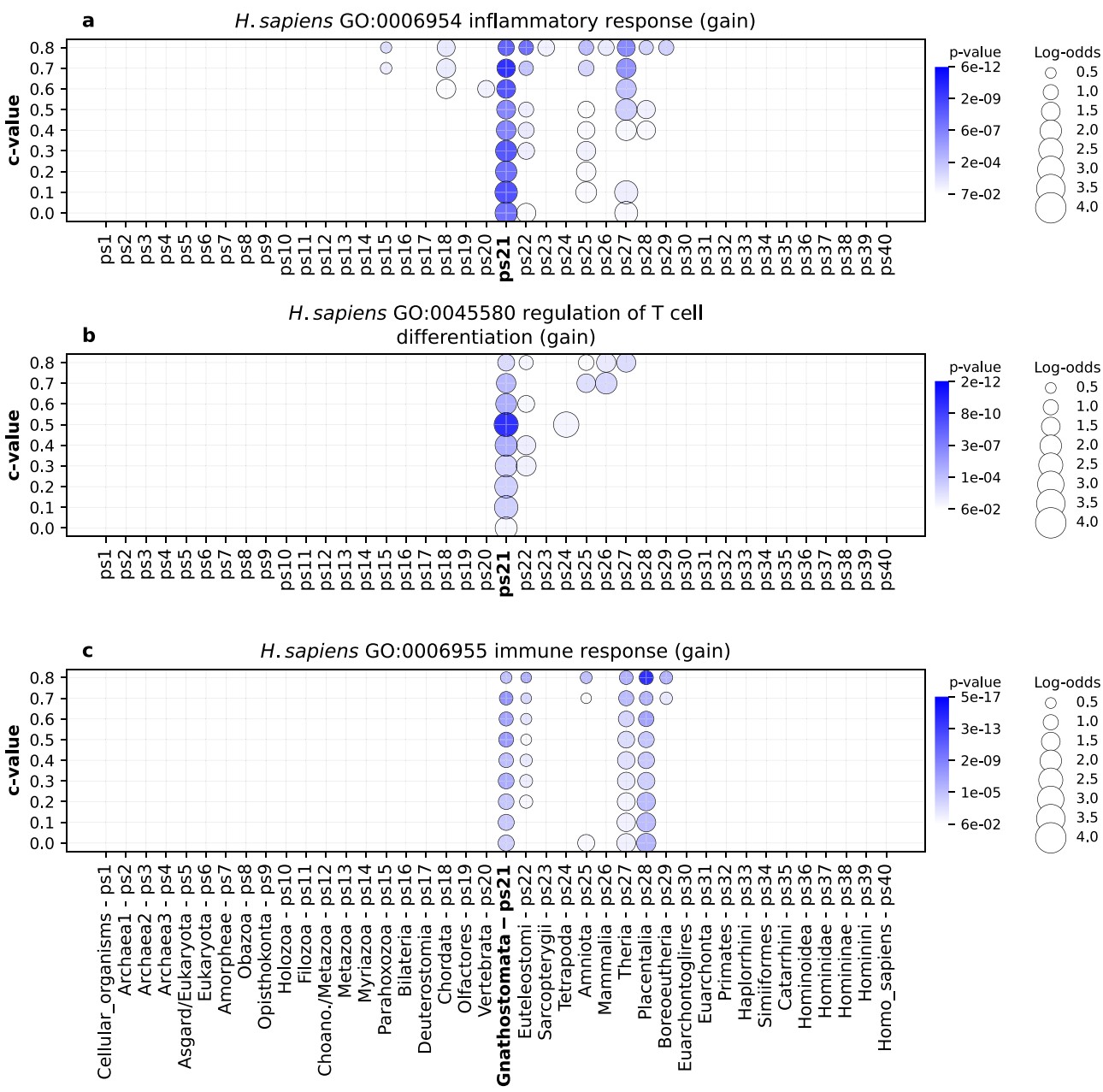

**Fig. 6 | The enrichment of GO functional categories related to adaptive immune system (*H. sapiens*).** The results are shown for three GO terms: **a** GO:0006954 (inflammatory response) **b** GO:0045580 (regulation of T cell differentiation), and **c** GO:006955 (immune response). Functional enrichments were calculated using the set of gene families gained along the *H. sapiens* lineage (x-axis). The gene families are reconstructed with MMSeqs2 *cluster* using a range of c values (0–0.8, y-axis). Solid circles depict significant enrichments of a GO term in gene families gained at a particular phylostratum. The size of circles is proportional to enrichment values estimated by log-odds, while the shades of blue correspond to *p* values. The significance of enrichments was estimated by two-tailed hypergeometric test corrected for multiple comparisons. Only enrichments with *p* < 0.05 are shown. These three terms show the strongest signal at the origin of jawed vertebrates (Gnathostomata, ps20), a group which evolved adaptive immune system. Other immune system related GO terms show similar patterns in the *H. sapiens* lineage (Supplementary Data 6). The source data of this figure are provided in the Source Data file.

To contribute to this idea, we here propose that complexity of an organism might be estimated by the number of conserved functions that are hardcoded in its genome (e.g., gene families) plus the number of functional benefits achieved through the direct biological interactions with other organisms (e.g., gene families in interacting organisms that contribute to that benefit). This way of viewing organism complexity goes beyond the holobiont paradigm[71] and moves the focus from the host and its microbes to the complete ecological community that interacts with a particular organism. In this respect, it would be important to distinguish between functions that are lost and are not needed any more, and those that are lost but outsourced, as only the latter contribute to organismal complexity. For instance, the loss of essential amino acids synthesis capability in metazoans[11] is compensated by the digestion of other organisms or through gut microbiota symbiosis. In this case the complexity score would not change, since other organisms produce essential amino acids for metazoans, which harness them through biological interactions. However, an opposite example would be the loss of cilia in fungi[72], because this functionality is not outsourced; the function is simply lost and not needed anymore in the fungal lifestyle.

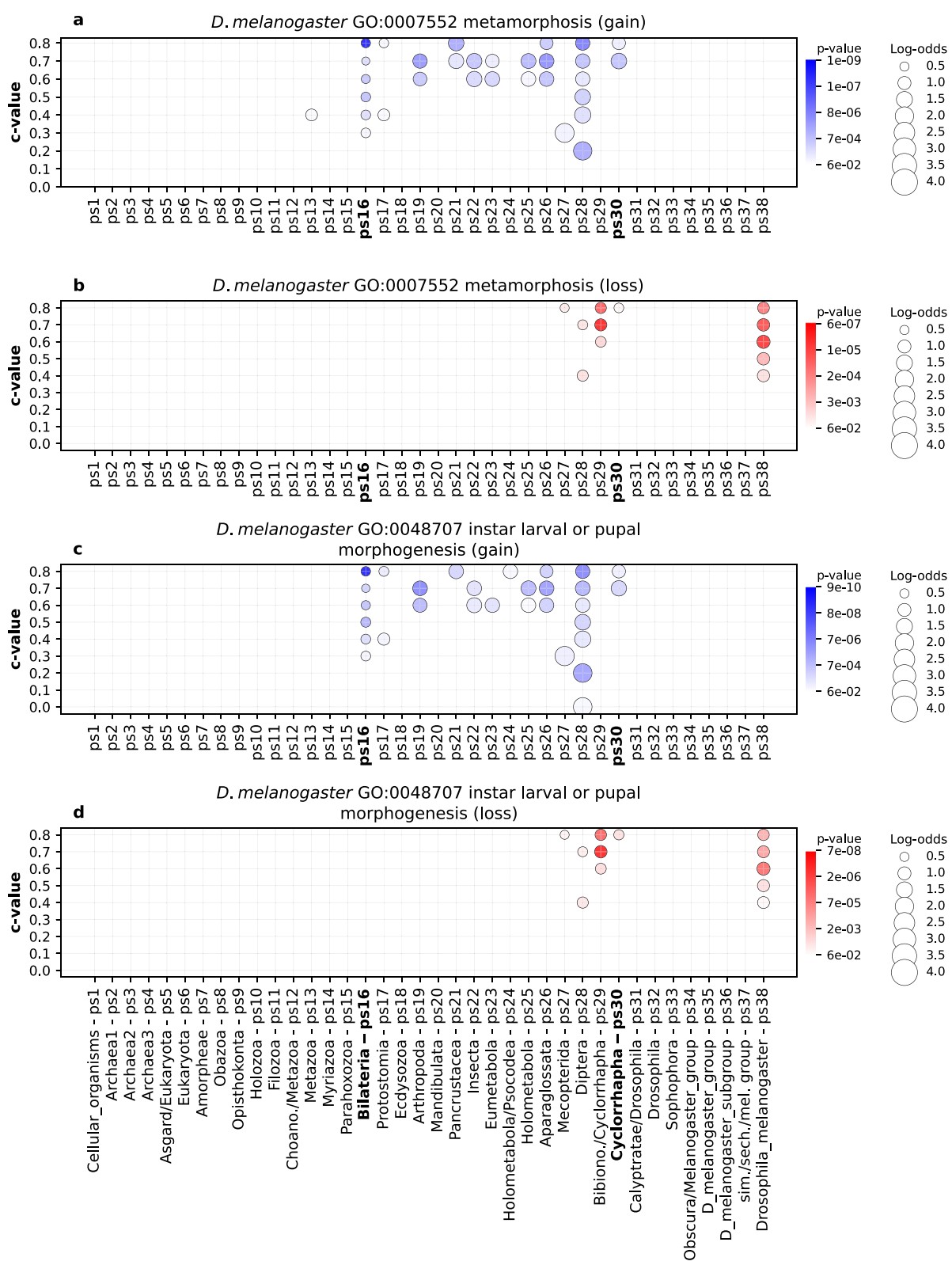

If one extends this reasoning further to fungi, then the very strong reductive trend in protein family diversity, which started already at Holomycota (ps10, Fig. 1c), suggests that fungi very early adopted lifestyles that include biological interactions, which allowed them to reduce the pack of protein families necessary for survival. Indeed, fungi evolved most likely from predatory protists that switched to parasitism[72]. Later, in the course of their evolution, they diversified by

making numerous ecological transitions to predatory, pathogenic, parasitic, and symbiotic interactions[72], all of which allow ecological specializations and lower the need for self-production of many protein families.

However, the pattern in plants is more intricate (Fig. 1d, Supplementary Data 3). After eukaryogenesis was completed (ps6, Fig. 1d), there is an obvious simplification trend in the lineage leading to plants

**Fig. 7 | The enrichment of GO functional categories related to metamorphosis in insects (*D. melanogaster*).** The enrichment profiles are shown for two GO terms: GO:0007552 (metamorphosis) and GO:0045580 (instar larval or pupal morphogenesis). The functional enrichments were calculated using the sets of gene families gained (**a, c**) and lost (**b, d**) along the *D. melanogaster* lineage (x-axis). The gene families are reconstructed with MMseqs2 *cluster* using a range of *c* values (0–0.8, y-axis). Solid circles depict significant enrichments of a GO term in gained or lost gene families at a particular phylostratum. The size of circles is proportional to

enrichment values estimated by log-odds, while the shades of blue (gain) and red (loss) correspond to *p* values. The significance of enrichments was estimated by two-tailed hypergeometric test corrected for multiple comparisons. Only enrichments with $p < 0.05$ are shown. These two GO terms show the strongest gain-signals in the range between the origin Bilateria (ps16) and the origin of Cyclorrhapha flies (ps30). Significant gene loss events are evident along diversification of dipterans (ps28-ps38). The source data of this figure are provided in the Source Data file.

with the lowest values at the origin of Streptophyta (ps11, Fig. 1d). This reduction in gene family diversity suggests increasing ecological complexification of aquatic ecosystems where these organisms thrived[63,73]. Nevertheless, after the colonization of land begun with Embryophyta (ps13, Fig. 1d), an opposite trend of increase in gene family diversity started, which finally reached the peak at the origin of Eurosids (ps23, Fig. 1d). This increase probably reflects new adaptations, via adaptive recruitment of new gene families to harsh conditions that plants had to face during challenging transition to land[73,74]. Finally, a strong simplification trend is evident along the diversification of Eurosids (ps24-ps32, Fig. 1d), which suggests an increase in biological interactions that allowed gene family diversity reduction. Indeed, the rapid diversification of Rosids that started in Cretaceous period resulted in the formation of rosid-dominated angiosperm forests present today[75]. This rosid radiation included nutrient and habitat specializations as well as coevolution with animals, especially insects and mammals[75]. All of this could have created conditions for gene family diversity reduction. A striking example in angiosperms that demonstrates the impact of nutrient and habitat specializations on the genome content are carnivorous plants, whose genomes show massive gene loss connected to functional outsourcing[76,77].

Our concept of functional outsourcing could be applied to parasite–host interactions as well. An excellent example are microsporidians—fungi-related eukaryotic intracellular parasites of animals. The genomes of microsporidians underwent an extreme reduction due to obligate intracellular parasitic life style[78,79]. However, not all lost molecular functions of microsporidians contribute equally to their genome complexity if viewed from the lens of functional outsourcing. Microsporidians lost mitochondrial respiration, including mitochondrial genome; however, they evolved adaptations for harvesting ATPs produced by the mitochondria of host cells[78–80]. This is a clear example of functional outsourcing, where an organism reduces a part of its genome but retains its benefits through a biological (parasitic) interaction. This interaction secures that the microsporidian ancestor, which possessed mitochondria, and the extant microsporidians that lack them have comparable complexity related to this particular function. An opposite example is the reduction in the number and diversity of splicing-related genes in some extant microsporidians[78,79]. These microsporidians compacted their genome and simplified their introns. As a result, a substantial part of the splicing machinery became obsolete and they lost it over time[78,79]. However, this functionality was not outsourced because microsporidians do not rely on the splicing machinery of host cells. This means that the overall complexity of microsporidians related to splicing decreased.

From the technical side, our analysis also uncovered that the minimal alignment length, as controlled by *c* value, is an important parameter that modulates the amount of macroevolutionary enrichment signal, which could be recovered on the gain-and-loss phylostratigraphic maps (Fig. 10, Supplementary Data 14). We quantified the amount of recoverable macroevolutionary signal by calculating the overall frequency of significant GO functional enrichments. In general, higher *c* values, which impose increasingly more stringent criteria on the sequence architecture, return the higher frequency of significant GO functional enrichments compared to the lower *c* values, which tend to return the lower frequency of significant functional enrichments (Fig. 10, Supplementary Data 14). However, in some instances lower

*c* values provide evolutionary informative patterns that are not visible at higher *c* values (e.g., ps6 Fig. 5a, b). We thus propose that, instead of choosing one *c* value cut-off, the best strategy is to explore protein sequence space using a broad range of *c* values and then inclusively evaluate evolutionary signals at hand.

Our result that higher *c* values typically return more statistically significant functional enrichments is also relevant in the context of debate on the importance of deep homologs for macroevolutionary reconstruction and the ability of sequence similarity search algorithms to detect them[33,36,81,82]. Our study clearly showed that functional information recovered by the enrichment analysis increasingly erodes in remote homologs (Fig. 10, Supplementary Data 14, Supplementary Data 6–13), which makes them less useful in tracking functional evolution. This effect is most likely underpinned by several factors. The most obvious one is that the generally lower number of clusters in permissive clustering (*c* value 0) lowers the power of hypergeometric test, which in turn decreases the number of significantly enriched GO categories. However, this factor must also be coupled with the cluster-size distribution change under permissive clustering (*c* value 0), where low *c* values are pushing sequences into mega-clusters mainly at the expense of moderately sized clusters (Supplementary Fig. 12). In addition, it is evident that clustering with the *c* value of 0 decreases the percentage of GO annotations in small to middle size clusters (Supplementary Fig. 13). This is probably yet another reason why we recovered in the permissive clustering (*c* value 0) less significantly enriched GO categories (Fig. 10).

However, studies that use sequence divergence simulations in an attempt to challenge macroevolutionary patterns obtained using real datasets, do not consider how the sequence divergence of artificially evolved sequences translates to their functional divergence[81,82]. This brings into question the biological relevance of these simulation studies, and probably explains why an attempt to link simulated sequences with functional evolutionary patterns failed[33,81,83]. Namely, it was suggested that the phylogenetic tracking of simulated sequences can produce false, but statistically significant, functional enrichments related to the evolution of germ layers in *Drosophila*[81]. This effect was initially attributed to the technical noise coming from the limitations of sequence similarity search tools. However, these simulation results turned out to be flawed due to a calculation/programing error[83].

Another problem related to the current simulation studies is that the underlying models assume that the genes evolve only through speciation and subsequent divergence from the starting sequence. This speciation-divergence model ignores gene duplications[1] and de novo evolution of genes[2,4,84,85] and thus its default expectation is that all simulated genes should be grouped as orthologs to the oldest phylogenetic node, irrespective of their sequence similarity or lack of it[81,82]. If this model is extrapolated to real sequences, this would mean that all extant genes evolved through speciation-divergence mode, and are thus deep orthologs that coalesce to some primordial sequences in LUCA, which is not realistic. This obviously ignores, alongside de novo gene emergence and gene duplications, the plethora of selective pressures that shaped sequence divergence of these genes over macroevolutionary time. In consequence, by insisting on grouping genes as deep orthologs (homologs), regardless of their sequence similarity, the current simulation models are largely uninformative in terms of reconstructing functional evolution.

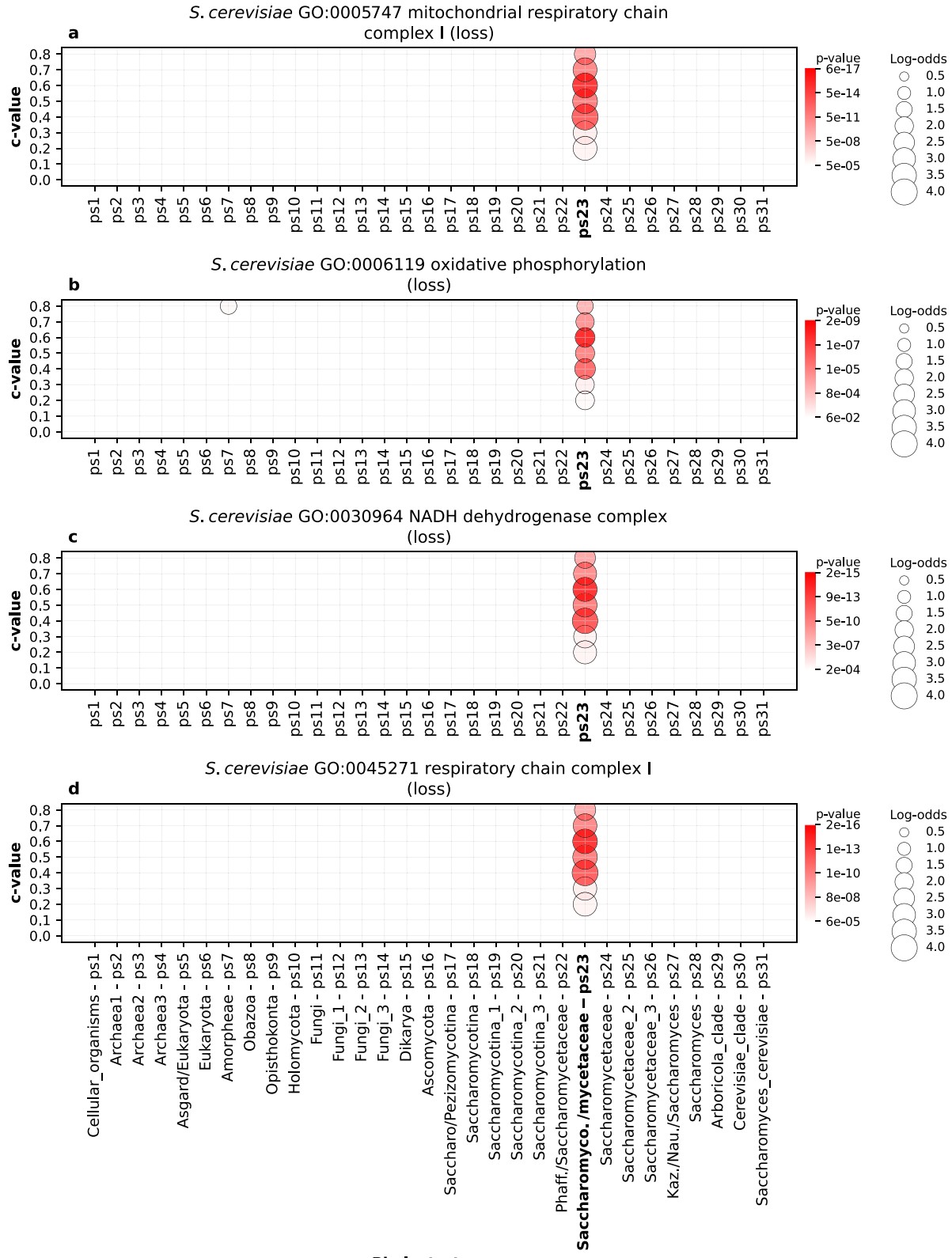

Yet another important aspect of our work relates to the benchmarking of computationally recovered gene family clusters in terms of their evolutionary and biological validity. There is a general consensus on the evolutionary origin of some biological features such as eukaryogenesis related innovations, adaptive immune system in animals, plastid endosymbiosis or the loss of respiratory chain complex I in yeasts. These and similar features could then be used to evaluate and calibrate bioinformatic pipelines with an aim to maximize the biological information that could be recovered by statistical analysis, e.g., functional enrichment analysis. This will also allow us to contrast the biological information content in gene clusters generated via different types of grouping strategies (e.g., orthologs vs homologs).

There are also some caveats to our approach. In our large reference dataset genomes are inevitably annotated with a heterogenous

**Fig. 8 | The enrichment of GO functional categories related to the loss of respiratory chain complex I (*S. cerevisiae*).** The results are shown for four GO terms: **a** GO:0005747 (mitochondrial respiratory chain complex I) **b** GO:0006119 (oxidative phosphorylation), and **c** GO:0030964 (NADH dehydrogenase complex) **d** GO:0045271 (respiratory chain complex I). Functional enrichments were calculated using the set of lost gene families along the *S. cerevisiae* lineage (x-axis). The gene families were reconstructed with MMseqs2 *cluster* using a range of *c* values (0 to 0.8, y-axis). Solid circles depict significant enrichments of a GO term in gene families lost at a particular phylostratum. The size of circles is proportional to enrichment values estimated by log-odds, while the shades of red correspond to *p* values. The significance of enrichments was estimated by two-tailed hypergeometric test corrected for multiple comparisons. Only enrichments with *p* < 0.05 are shown. These four terms show the strongest loss signal at the origin of Saccharomycodaceae/Saccharomycetaceae (ps22), a phylostratum (Supplementary Data 1) that lost respiratory chain complex I. The source data of this figure are provided in the Source Data file.

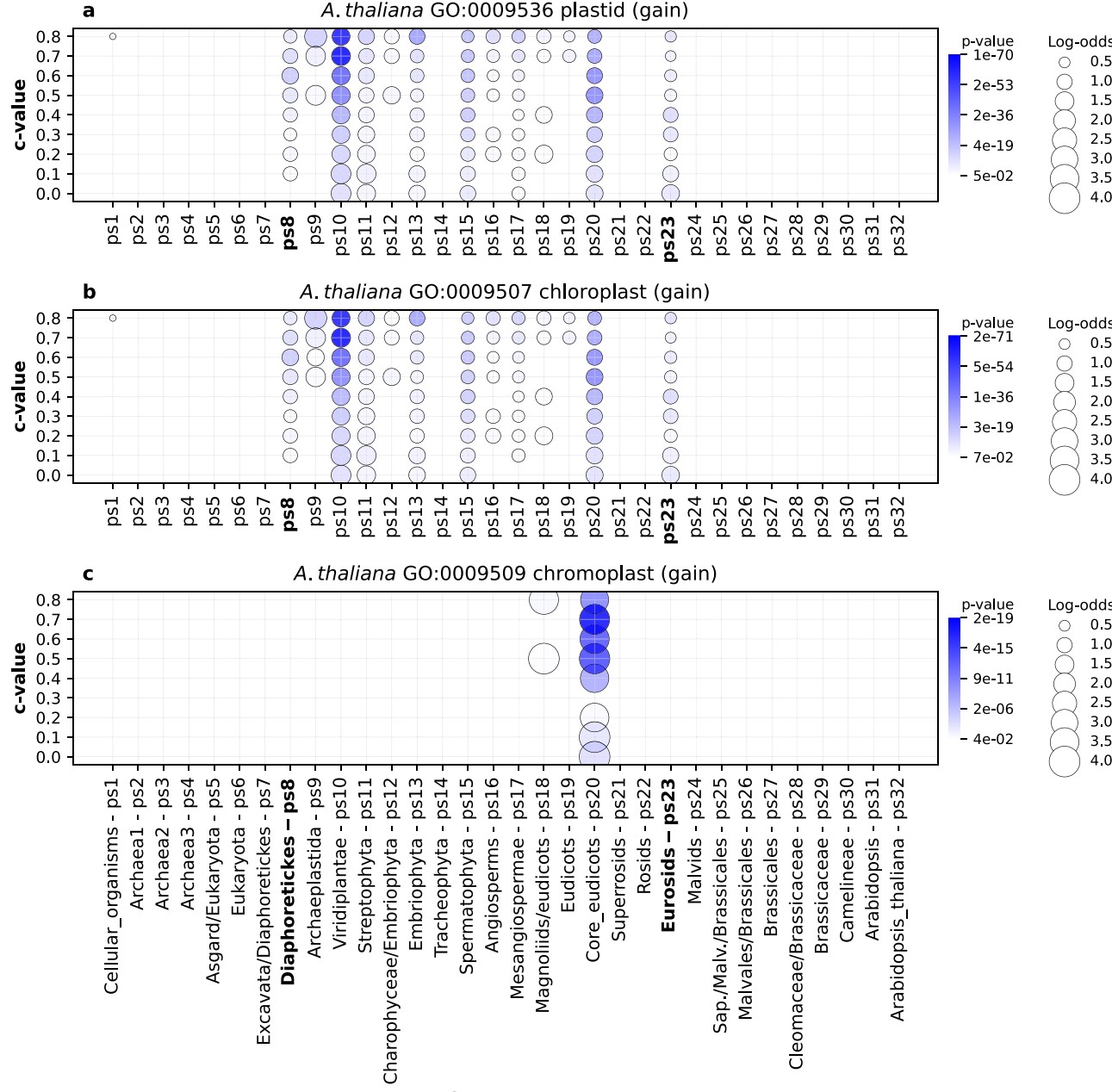

**Fig. 9 | The enrichment of GO functional categories related to plastid evolution (*A. thaliana*).** The results are shown for three GO terms: **a** GO:0009536 (plastid) **b** GO:0009507 (chloroplast), and **c** GO:0009509 (chromoplast). Functional enrichments were calculated using the set of gene families gained along the *A. thaliana* lineage (x-axis). The gene families are reconstructed with MMseqs2 *cluster* using a range of *c* values (0–0.8, y-axis). Solid circles depict significant enrichments of a GO term in gene families gained at a particular phylostratum. The size of circles is proportional to enrichment values estimated by log-odds, while the shades of blue correspond to *p* values. The significance of enrichments was estimated by two-tailed hypergeometric test corrected for multiple comparisons. Only enrichments with *p* < 0.05 are shown. These three terms show the strongest loss signal in the range between Diaphoretickes and Eurosids (ps8-ps23). A weak, but significant signal that reflects the origin of plasmids from cyanobacteria is evident at the origin of Cellular organisms (ps1, **a, b**). The source data of this figure are provided in the Source Data file.

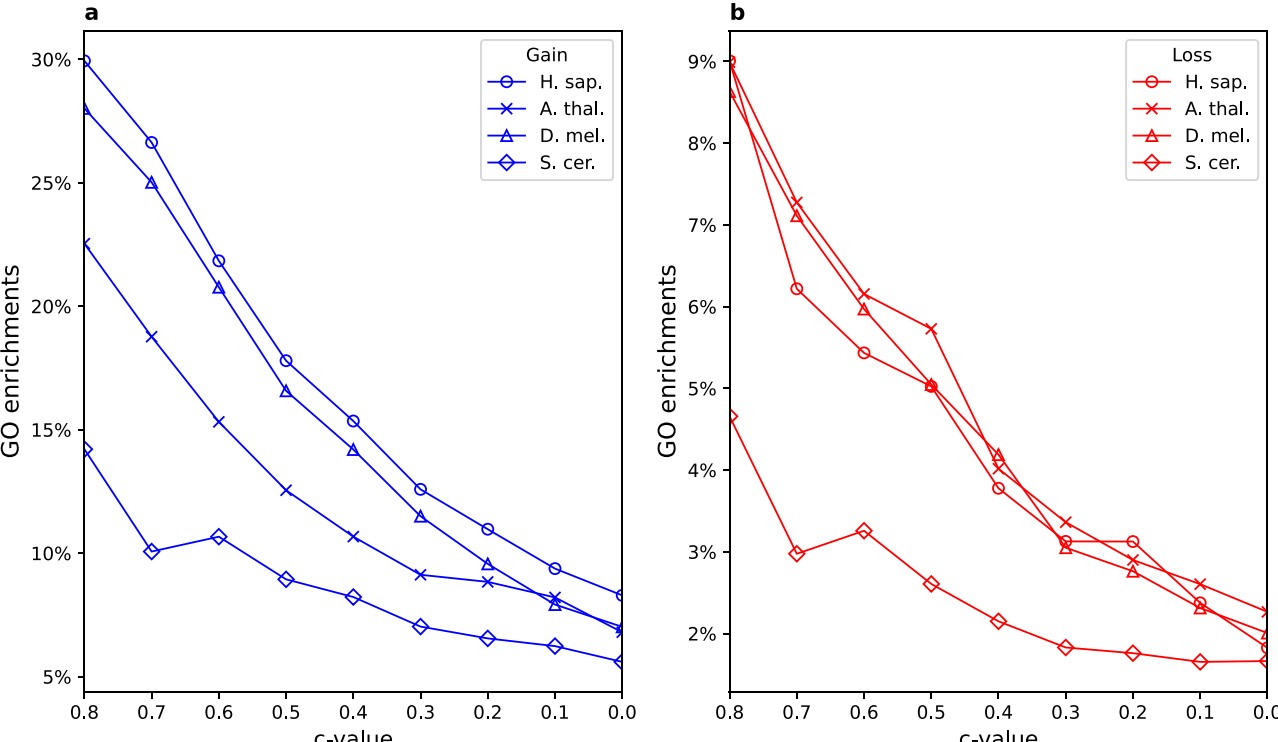

**Fig. 10 | The overall frequency of significant functional enrichments (GO) is dependent on *c* value.** We estimated the amount of recoverable macroevolutionary signal by calculating the overall frequency (percentage) of significant GO functional enrichments among (**a**) gained and (**b**) lost protein families (Supplementary Data 14). A GO function was counted as enriched if we detected at least one phylostratum to show significant enrichment for the given function, for particular *c* value and focal lineage. To obtain GO enrichment frequencies we divided the number of significantly enriched GO functions with the total number of tested GO functions. The trends are shown for the four focal lineages (*H. sapiens*,

*D. melanogaster*, *S. cerevisiae* and *A. thaliana*). The significance of GO functional enrichments was estimated by two-tailed hypergeometric test corrected for multiple comparisons. Only enrichments with $p < 0.05$ are considered significant. The gene clusters (protein families) are reconstructed with MMseqs2 *cluster* using a range of *c* values (0–0.8, *x* axis). In general, higher *c* values, which impose stringent criteria on the sequence architecture, return the increasingly higher frequency of significant enrichments compared to the lower *c* values. The source data of this figure are provided in the Source Data file.

set of annotation pipelines. This annotation heterogeneity can cause that many genes erroneously appear lineage-specific[86]. However, this effect has stronger impact on relatively closely related species, whereas in distantly related organisms as well as in the large sets of genomes this effect should be less pronounced[86]. Consequently, we expect that in our analysis annotation heterogeneity might have some impact on the youngest phylostrata, but not so much on the deeper phylogenetic nodes. Similarly, horizontal gene transfer (HGT) can pull the origin of gene families to older nodes on the phylogeny, although not to such extent as genome contaminations, which we carefully controlled. However, HGT in eukaryotes is generally less frequent then in prokaryotes[87,88], thus these effects should have lower impact on the eukaryotic part of the phylogeny. In addition, the impact of HGT will also depend on the phylogenetic distribution of a gene family. For instance, the estimated phylogenetic origin of a gene family whose members are broadly taxonomically distributed will be less sensitive to HGT than the estimated phylogenetic origin of a gene family which is specific for a narrow lineage.

In conclusion, our study demonstrates that the patterns of gene family gain and loss correlate with major evolutionary and ecological transitions. It seems that the gene family loss follows the evolutionary radiations of major multicellular eukaryotic lineages as a consequence of ecological complexification that allowed niche specializations and functional outsourcing. This in turn suggests that in evaluating the complexity of an organism, in addition to the number of its conserved parts (e.g., gene families), one should also consider its biological interactions and the functional context that sustain its existence.

## Methods

### Consensus phylogeny

Using information from the relevant phylogenetic literature[75,89–95] (for the full list of considered studies see Supplementary References), we constructed a consensus tree with 667 species taxa, whose genomes were publicly available (Supplementary Data 1). In the assembly of the tree, we aimed to comprehensively cover lineages that lead from the origin of cellular organism to our main focal species; i.e., *H. sapiens*, *D. melanogaster*, *S. cerevisiae* and *A. thaliana*. We chose these main focal species because they are well-studied model organisms, which allowed us to reach an adequate number of phylogenetic levels (phylostrata) and to populate them with the suitable amount of reference genomes. In addition, these lineages represent the evolutionary trajectories of deuterostomic animals, protostomic animals, fungi and plants, all of which are ecologically important and evolutionary successful multicellular eukaryotic groups with currently the best annotation of gene functions.

### Reference genomes

We retrieved the full protein sequence sets (reference genomes) of our 667 taxa mostly from the ENSEMBL and NCBI databases. In addition, we retrieved some reference genomes, which were not available in these databases, from other sources (Supplementary Data 16). In every reference genome, we retained only protein sequences that come from the longest splicing variant of a respective gene and that are capable to return a self-hit in sequence similarity searches. To find the protein sequences without self-hits, we compared each sequence to itself using MMseqs2 (version 14-7e284) *easy-search* with −e 0.001.

To evaluate the quality reference genomes, we used BUSCO v5.4.3 with default parameters[96]. We analyzed all reference genomes by using six BUSCO lineage datasets independently: bacteria_odb10, archaea_odb10, eukaryota_odb10, metazoa_odb10 viridiplantae_odb10 and fungi_odb10[97]. In order to detect possible contaminations, which might produce systematic biases by erroneously assigning protein families to non-native ancestral nodes[98], we looked for genomes that had comparably high BUSCO scores in lineages other than the native one. After detecting such genomes, we substituted them with better versions, either coming from the same or closely related species. In the next step, we individually evaluated every genome that had BUSCO duplication score above 20%, with an aim to detect eventual within-lineage contaminations. In some cases, high duplication scores are expected because of known duplications events. However, in other cases we substituted these genomes with better versions that have lower duplications scores if such versions were available. After this multistep contamination cleanup, we further evaluated genomes that had BUSCO completeness scores within native lineage below 80%. If possible, we substituted these with better versions, either coming from the same or closely related species. However, we also retained some of these genomes because their relatively lower completeness scores reflect their derived biology or parasitic lifestyle and not technical issues[97]. For instance, *Encephalitozoon cuniculi* and other microsporidians have very low completeness scores due to their parasitic lifestyle which led to severe reduction of their genomes. In sum, our set of reference genomes had an average completeness score more than 90%. The list of reference genomes and all their corresponding BUSCO scores across six BUSCO lineage datasets is available in Supplementary Data 16.

## Sequence similarity search and clustering

The protein sequences of all reference genomes were clustered using the *mmseqs cluster* algorithm within the MMseqs2 package (version 14-7e284), with the following parameters: *-e 0.001, -s 4.0, --max-seqs 400, --cluster-mode 1* and *--cov-mode 0*[23,24]. This algorithm integrates all-against-all protein sequence similarity search with clustering procedure. We retained the MMseqs2 *cluster* default e-value cutoff of $10^{-3}$ as this threshold was repeatedly shown to be optimal[1,4,33,36]. To cluster our set of protein sequences, we applied connected component algorithm (*--cluster-mode 1*). In contrast to other available clustering options in MMseqs2 *cluster*[25], this is a transitivity-based clustering algorithm that forms clusters with more remote homologs. We independently clustered our protein sequence set nine times by varying *c* values in the range between 0 and 0.8 with 0.1 increments. A *c* value determines the minimal percentage of aligned sequence length to a representative sequence (0–80%) that is required for the assignment of a protein sequence to a cluster. The alignment coverage was calculated under *--cov-mode 0* which calculates the percentage of alignment overlap in reference to a longer protein sequence. The higher percentages of alignment coverage ensure that all sequences within a cluster share increasingly similar overall sequence architecture (stringent criteria). In contrast, lower *c* values permit clustering based on the conservation of shorter protein sequence stretches irrespective of their arrangement within genes (permissive criteria). This approach allowed us to explore evolutionary relevant shifts in the protein sequence space at the level of short conserved protein sequence stretches (*c* value 0), at one extreme, and at the level of sequence conservation along almost complete gene length (*c* value 0.8) at the other one.

In contrast to previous studies that rely on orthology-focused clustering that filters out a part of sequence similarity information[11,13,14], e.g., by considering only the reciprocal best hits[13,21], our clustering protocol takes into account all detectable sequence similarity below an *e* value cut-off and above a *c* value threshold to form clusters. This approach allowed us to unrestrictedly explore protein sequence space

and determine significant shifts in the form of gene family gains and losses. We previously repeatedly showed that these detectable jumps in the protein sequence space carry biological information, which can be then statistically recovered[1,29–31,33,36].

To compare the MMseqs2 *cluster* algorithm and an MCL-based approach, we first made all-against-all sequence similarity comparison using the MMseqs2 *search* algorithm[24]. This is an independent tool within the MMseqs2 package (version 14-7e284) that has BLAST-like behavior in the terms of sensitivity, but much higher computation speed[24]. The input to the MCL clustering was a distance matrix, where each distance (weight) was the negative logarithm of the e-value for a sequence pair. An e-value less than 1e-200 was set to a maximum value of 200, i.e., the maximal edge weight was 200. Since the *e* values may differ for a pair of sequences depending which sequence was used as a query, we used the better value between the two. These options were applied following suggested MCL protocols for clustering graphs of protein sequences[28,99]. In the next step we clustered the distance matrix using the MCL (version 22-282) algorithm in two independent runs, with inflation parameters 2 and 1.5, respectively[28]. In the final test, we added to our MCL pipeline a gene length and phylogenetic distance normalization procedure as described in the OrthoFinder paper[42]. The bit-scores obtained by all-against-all sequence similarity comparison using MMseqs2 *search* were normalized using the OrthoFinder approach which we implemented in our custom-made python script (normBitScore.py). Finally, we used the obtained distance matrix with normalized bit-scores as an input to the MCL algorithm in two independent runs, with inflation parameters 2 and 1.5, respectively.

## Gene family gain and loss reconstruction

By referring to our 667 reference genomes we determined the taxonomic composition of every cluster (gene family) obtained by the MMseqs2 *cluster* algorithm. For every lineage that leads from the root of the consensus tree to one of our 667 species, we extracted gene families that appear along that particular lineage. In this way, we obtained 667 sets of gene families which are relevant for a particular focal lineage. We retained only those gene families in the terminal (youngest) phylostrata that contain at least two cluster members. By applying Dollo's parsimony, we then determined for every gene family that appears along a specific lineage the phylostratum of its gain and eventually of its loss. To obtain a gene family gain and loss profile for the whole tree, we made the union of all gain and loss events along 667 lineages. Using this cumulative gene family gain-and-loss information, we calculated gene family content in every node on the consensus tree (Supplementary Data 1, 2, 3, Fig. 1). It is important to note that a gene family gain event must always precede its loss by at least one internode (i.e., two phylostrata). This simply reflects the fact that a gain of a gene family in a particular phylostratum and its loss immediately in the next one excludes that gene family from the focal lineage and makes it specific for the respective side branch. For this reason, the loss events in a particular focal lineage, and in the whole tree, could be calculated the earliest at the third phylostratum from the root.

## Functional enrichment analysis

We functionally annotated protein sequences in the reference genomes with COG and Gene Ontology (GO) annotations using *emapper* version 2.1.9, the database version 5.0.2 and *diamond* version 2.0.15 search algorithm within the EggNOG tool[100]. To functionally annotate clusters (gene families) with either COG or GO terms, we applied a simple rule that every functional annotation of a protein member within a cluster is also assigned to the whole cluster. This strategy allowed us to preserve available functional information, which for some clusters and taxa was rather limited.

However, we also tested a more stringent criteria by setting up a threshold that a functional annotation is transferred to a cluster only if at least 50% of the genes within the cluster are annotated with a particular function. This stringent procedure, by lowering the number of COG and GO annotations, reduced the power of hypergeometric test. As a result, the total number of enriched functions dropped, especially affecting enrichments at lower $c$ values. Due to their granularity, this effect was more pronounced for GO functions where we recovered the 35% of initial enrichments. Nevertheless, the recovered enrichment patterns under this stringent criterion were not qualitatively different from those obtained in the initial procedure for COG and GO annotations (Supplementary Fig. 14, Supplementary Data 17, 18), so we chose to present the results of the initial procedure which preserves functional information. To calculate a COG or GO term functional enrichments in gained or lost gene family sets along focal species lineages, we performed two-tailed hypergeometric test for each focal species and $c$ value independently. To correct for multiple testing, we adjusted $p$ values using Benjamini–Hochberg method as implemented in the Python *statsmodels* library.

The aim of these functional enrichment analyses was to detect lineage-specific functional adaptations. To achieve this, we looked at a functional annotation along a particular lineage and then discerned phylogenetic nodes that have unusually high concentration of that function. For every focal lineage independently, we compared the frequency of a COG or GO function in a phylostratum to the overall frequency of that function in the focal lineage. In this type of analysis, we took only a focal lineage as a reference to avoid collating the functional signals of many parallel evolutionary events, which would consequently obscure lineage specific adaptations. In this context, it is expected behavior that at a node shared between multiple lineages an enrichment signal varies depending on a focal lineage. This is evident in the enrichment analysis of GO term "cilium organization" (Fig. 5). Although there is always the same number of cilium-related functional annotation at the origin of eukaryotes (ps6, Fig. 5), whether they are enriched or not will depend on the overall distribution of this functional annotation along all nodes that define a specific focal lineage. For instance, in the yeast analysis we detected an enrichment signal at the origin of eukaryotes (ps6, Fig. 5c). This enrichment signal is quite strong since in the later fungi-specific nodes there is no massive innovations related to cilium functions that would overshadow the signal at ps6. Moreover, we correctly detected that later on in the fungal linage cilium-related functionality is entirely lost (Supplementary Data 12). In contrast, animal lineage experienced a massive gain of cilium-related functions at choanozoans (ps12, Fig. 5a, b), which is a statistically more surprising event than their previous gain at origin of eukaryotes (ps6, Fig. 5a, b).

### Reporting summary

Further information on research design is available in the Nature Portfolio Reporting Summary linked to this article.

## Data availability

The authors declare that the data supporting the findings of this study are available within supplementary information files and in Figshare[101] at https://doi.org/10.6084/m9.figshare.20522103.v1. Accession codes of sequence data analyzed in this study are listed in Supplementary Data 16. Source Data for Supplementary Fig. 6 can be found in Supplementary Data 15. The source data of all other display items are also provided as a Source Data file. Source data are provided with this paper.

## Code availability

The custom made code used in this study is available in GitHub at https://github.com/PhyLoss/PhyLoss and at Zenodo[102].

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

## Acknowledgements

We thank M. Futo, S. Koska, N. Čorak, and D. Kifer for discussions. This work was supported by the Croatian Science Foundation under the project IP-2016-06-5924 (T.D.-L.), the City of Zagreb (T.D.-L.), the Adris Foundation (T.D.-L.) and the European Regional Development Fund KK.01.1.1.01.0009 DATACROSS (M.D.-L., T.Š., and T.D.-L.). We used the computational resources of the University Computing Center -SRCE (Isabella), and the Ruđer Bošković Institute.

## Author contributions

T.D.-L. initiated the study, T.D.-L. and M.D.-L. conceptualized the study and constructed the phylogeny. M.D.-L. and K.Š. prepared the genomic data, M.D.-L. developed the phylostratigraphic pipeline and the algorithm for the computation of gene family gain and loss. M.D.-L. and T.Š. performed functional annotations of clusters. T.Š. and M.D.-L. wrote the scripts for the functional analysis. M.D.-L., T.Š. and K.Š. wrote the scripts for the cluster analysis. All authors analyzed the data. M.D.-L., T.Š. and K.Š. prepared the figures and tables for publication. M.D.-L. and T.D.-L. wrote the manuscript. All authors read and approved the manuscript.

## Competing interests

The authors declare no competing interests.
