## [Peer Review File · Nature Communications]

Macroevolutionary dynamics of gene family gain and loss along multicellular eukaryotic lineagesREVIEWER COMMENTS

Reviewer #1 (Remarks to the Author):

I had already read a previous submission of the paper titled "Macroevolutionary dynamics of gene family gain and loss along multicellular eukaryotic lineages" by Domazet-Lošo and collaborators. I raised a series of concerns in the previous submission, that are not addressed in this one. I will share here my previous comments (some line numbers might be off).

This study analyses the 667 complete genomes across the Tree of Life to profile the gene complements of different ancestors, going back to the last common ancestor of cellular life. The analyses show ancestral peaks of genome complexity followed by simplification, used to discuss the complexity by subtraction and the biphasic models of genome evolution, as well as showing that new genes display functional enrichment while functions gene losses seem to be evenly spread.

However, there is a major methodological concern that casts major doubts on the paper's findings. The comparative genomic analyses are anchored in four focal species, instead of performing evolutionary analyses in which all the species analysed are considered together. This approach is not up to the standards of the field, and it is reminiscent of phylostratigraphic approaches that have been often criticised in the literature. Focus on the genes of one species means that all the patterns of gains and losses specific to this lineage, from the origin of life to nowadays, are going to severely impact the reconstruction of the gene complements in ancestral nodes. For example, using the genes present in *Homo sapiens* to reconstruct the gene complement of the first bilaterian means that all the genes lost in the nodes for deuterostomes, chordata, olfactores, etc. will be omitted from the reconstruction of the bilaterian ancestral genome.

Proof of this is found in the same paper. For example, one of the figures shows the presence of cilium-related genes in the first eukaryote (LECA), and the number of these genes in LECA changes depending on the focal organisms used. This makes no biological sense. How many other genes have been missed due to only looking at these four species? We can reconstruct the ancestral genome of the first eukaryote using all the genomes of eukaryotes at once, there is no reason to limit analyses to four terminals. Evolutionary studies have to consider all the taxa studied and compare them all vs all, otherwise, they run into biases and artefacts. In a time in which most papers map gene gains and losses in the nodes of a phylogeny, considering hundreds of species at once, inferred using sophisticated methods, it is not adequate to draw major conclusions about major evolutionary transitions from only four species. What is the scientific rationale for this artificial constriction?

Another major issue is the conflation of gene complements in different ancestors with gene gains and losses. For example, Figure 1 shows ancestral gene complements which are used to justify the trajectory of genomic gains/losses across the Tree of Life. But ancestral complements do not reflect gains and losses. For example, a given node could gain 1000 gene families and lose another 1000, and the number of genes in Figure 1 wouldn't change. This argument is used to support different hypotheses around the role of gains and losses in different parts of the Tree of Life.

Finally, at the methodological level, the authors criticise some of the current state-of-the-art methods (e.g., OrthoFinder) for not considering internal sequence conservation. Their approach using different c-values is interesting. However, they do not mention in their approaches any correction in the gene searches for gene length. The impact of gene length in BLAST searches is well-known, and important when looking at deep homology (e.g., older longer genes will be assigned to the correct gene family more often than shorter genes). Similarly, the specific topology used can have a dramatic impact on Dollo parsimony reconstruction. The one provide in Supp File 1 shows many collapsed nodes, which cannot possibly have been used to infer gains and losses. For example, how is it possible to infer which genes are new or lost to animals using Dollo parsimony if the phylogeny does not resolve which is the first-splitting lineage of animals? What about LUCA?

Some other comments that I hope are of help to the authors:

1. The introduction picks on a few recent studies on animal evolution but neglects to mention many other ones around plants, fungi, and LECA published in the last few years. These studies have been published in high-impact journals. This is more shocking in the results and discussion sections, where large conclusions are drawn around the genomic changes in these nodes with no reference to other studies that showed the same patterns before.
2. Likewise, the article raises an interesting point about the different levels of clustering used in different studies. However, it fails to recognise the levels used in the various articles and misassigns them. For example, line 80 claims that some of these studies use orthogroups when some of these articles actually do use homology groups and they are even named as such in the papers.
3. The study does not dwell on the last common ancestor of all cellular life. Figure 1 shows 8k-12k gene families in this ancestor, which means this ancestor of all life had at least that many genes if not more. This is possibly the most striking finding in the study but it is totally overlooked. Which are these gene families? How does this elevated number of gene families compare to other studies (e.g., recent papers by Bill Martin)?
4. L253, how is it possible to get a gain/loss ratio for LUCA in Figures 2 and 3 if there are no outgroups for LUCA? In which other organisms are the genes lost in LUCA found? Shall not be the gain/loss ratio infinite?
5. Different parts of the paper (e.g., L86, 572, 577, etc) claim that the approach used here recovers more "macroevolutionary changes" or more "biologically significant" findings than other approaches. Which criteria are used to make these claims and to identify how significant are those changes compared to other studies?
6. I disagree with the concept that all extant genes are homologous at some level, as all coalesce to the first gene that ever existed. That assumes that genes only emerge by gene duplication. This is surprising based on the pioneering work of the authors about de novo gene generation. Does it mean that de novo genes don't leave any descendants? It also ignores that new genes can emerge by gene fusion or fission.
7. Similarly, L500 links the simplification of bacterial genomes in pangenome contexts with genome simplification in multicellular eukaryotes due to diet changes. This relation is far-fetched and would require evidence to support it.
8. Related to the previous point, L525 suggests that organismal complexity could be estimated from the number of ancestral conserved functions plus the functions achieved through interactions with other organisms. How this applies to organisms such as parasites? A parasitic worm with no digestive tract or significant nervous system would be inferred as complex because most of the lost functions are performed via the host.
9. L583 and the following criticise studies using simulations (Moyers, Zhang, Paschalis) for their lack of functional meaning. While this point is correct, these papers never claimed to provide functional findings. Their point is to highlight methodological issues, and they are really great at doing so.
10. Methods claim to use BUSCO to filter genomes based on quality, however, there is no mention of the completeness threshold used to select such genomes.

Reviewer #2 (Remarks to the Author):

Review

In the manuscript "Macroevolutionary dynamics of gene family gain and loss along multicellular eukaryotic" by Domazet-Lošo et al., the authors infer putative orthology across a large sampling of Eukaryotic genomes, aiming to reconstruct the history of gene gain/loss patterns. This analysis focuses on the use of four focal species, which are well-known model organisms from animals, plants, and fungi. Although the authors examine the phylostratigraphy of additional taxa (e.g., the truffle), the use of such few 'focal' species to represent the evolutionary dynamics of entire kingdoms seems a bit misleading. I think that are great reasons to use these 'focal' species as controls, but insights into other lineages with interestingly evolutionary dynamics would bolster the impact of the manuscript.

Major comments

- 1) Line 20: To say four focal species represent the diversity of three kingdoms is a stretch. Could this be rephrased throughout?
- 2) As a control, does your method identify known whole genome duplication events, such as the ones in budding yeast or flowering plants?
- 3) Much of the gene family analysis will depend on taxon sampling. Can the authors provide a figure to detail the depth of taxon sampling at each ps-level?
- 4) More broadly, the meaning of the ps-levels isn't very clear. For example, are the steep increase and decreases due to a confounding of large geologic time between taxonomic ranks? Inversely, couldn't the lack of pronounced peaks at ps 23-35 in the phylostratum of *S. cerevisiae* be largely attributed to a lack of evolutionary time between each taxonomic rank? Although taxonomic ranks are informative, and a reasonable x-axis for figure 1, I think the time-component is overlooked and should somehow be accounted for, or at least discussed.
- 5) The authors did a nice job comparing MMseqs2 to MCL-based clustering. Advances in orthology inference — such as implemented in OrthoFinder — also account for length-related biases in sequence similarity scores. Not accounting for these biases results in oversplitting orthologous groups of genes. Does MMseqs2 also conduct this correction? If not, this may explain the larger number of gene families observed in, for example, Figure S3 and the root of Eukaryota.
- 6) The analysis of lineage-specific genes may be complicated by the use of different gene annotations predicted using different software. See Weisman et al. <https://www.sciencedirect.com/science/article/pii/S0960982222007217>. This may be worth noting in the discussion.
- 7) Can the authors provide greater detail about the Dollo parsimony-based algorithm employed for their analysis? It is not clear what was used in the GitHub repo to do that analysis.

Minor comments

- 1) Can the authors clarify what they mean by "if the quality looked compromised" on line 857?
- 2) Version information is provided for some, not all software. For reproducibility, can this information be included in the manuscript?

Reviewer #3 (Remarks to the Author):

Domazet-Lošo et al. have profiled the genomic content and macroevolutionary trajectories from the ancestor of cellular organisms to four model organisms (two animals, one plant and one fungus). To achieve sufficiently detailed clustering and high resolution for phylostratigraphy, they included 676 whole genomes in the analysis. The results are robust, which could be partly due to

this impressive dataset. The other reason could be the clustering approach, which explores a high range of possible scenarios without choosing an arbitrary threshold, even though the chosen clustering approach may be burdened with biases. Another indisputable advantage of the study is that they seek to extend the analysis to deeper nodes in the tree of life than ever before. They found that gene family content follows a common pattern between these different model species. The number of gene families reaches a high value (genomic complexity) from which we can observe a gradual decrease towards the extant species. Based on the current literature, this is the expected pattern, but the generality and robustness of the phenomenon presented here would be a significant contribution to the field. Furthermore, they provide exciting hypotheses for the existence of reductive evolution (ecological specialization and functional outsourcing), although these hypotheses are based mainly on the literature and not proven by their own results. The paper is carefully drafted, the approach and methodology is adequate but might necessitate some care/reconsideration. Based on the highlighted examples the COG and GO enrichment also seems to be reliable, but methodological questions are raised with this from my side.

Overall, I think this article would represent a considerable step in understanding the gene family evolution of eukaryotes. However, I see some potential methodological weaknesses that should be addressed during the revision process, in order to rule out the possibility of biases.

major points:

1. One of the most obvious advantages of the study is using a large number of genomes to determine the most accurate cluster membership and phylostratigraphic resolution. Nevertheless, the authors failed to exercise reasonable caution about the quality of the genomes used, which can greatly affect the analyses of genome evolution, especially the ancestral genome reconstruction. The contamination of genomes could lead to an overestimation of gene content in ancestors and inflates the rate of gene loss, as shown by Balint et al. *bioRxiv* 2022: <https://www.biorxiv.org/content/10.1101/2022.11.17.516887v1.full.pdf+html>). For example, the genome of *Quercus suber* contains an entire fungal genome as contamination, resulting in a >6000 (cca. 60%) copy number overestimation for the LECA in the study Balint et al. (2022). *Quercus suber* is also listed in the Supplementary Table 1 of Domazet-Lošo et al. Biased overestimation of LECA is consistent with the phenomena presented by the authors in Fig 1 (higher gene family content in ancient nodes and gradual loss towards the recent nodes). Therefore, I strongly recommend the authors to recheck/clean up the genomes used to exclude this bias, or at least get a picture about the extent of the bias.
2. Horizontal gene transfers and incorrect gene annotation (e.g. artificial fusion proteins especially at low c-values) can also influence the phylostratigraphy, although not to such an extent as genome contamination. HGT was mentioned once (L632-634), but these limitations of the approach should be emphasized more strongly, e.g. in the methods or in the discussion.
3. The clustering approach touches on extreme setups (connected component c-value=0), therefore the following descriptive statistics on clusters are worth to be prepared as graphs or tables and attached as supplementary files. 1) Distribution of cluster sizes, similar to Ocaña-Pallarès et al. (*Nature*, 2022 Supplementary Information 2-Fig. 7.). This would be useful for assessing the reconstruction of the ancestral genome of LECA (L293-307). 2) Domain and/or GO diversity of clusters, which would be useful for understanding the relationship between functional signal and alignment length/clustering strategies (Figure 10). I have the impression that simply the higher number of clusters in strict clustering (c-value=0.8) could lead to an increase in the number of significantly enriched GO categories. 3) Pairwise distribution of cluster coverage for each clustering setup (different c-values). A pairwise coverage threshold is not necessarily true for all components of a cluster. Therefore, the statement "c-value = 0.8 which forces at least 80% of sequence length alignment overlap" can be applied to the pairwise protein pairs but not to the final clusters.
5. Comparing clustering approaches (connected component vs MCL) is a useful and essential part of the work. However, as it stands, paragraph L293-307 is not so convincing, or partially unnecessary. In the first part of this paragraph, the authors argue that the gene family content of LECA was estimated to be 12,500 clusters (with the most permissive clustering, c-value=0), which number is close to the recently published estimation: 10,233 Pfam domains in LECA. Therefore, - they argued - the chosen clustering approach is better than MCL clustering which estimated 24-27,000 clusters. Since the authors have not shown/proven the domain homogeneity of the most permissive clustering (c-value=0) the two values (number of clusters and number of Pfam domains) should not be compared to each other. Also the authors state elsewhere that attraction of multidomain proteins in large clusters is an issue (L156-157). The second part of this paragraph

(L302-L307) deals with the computational speed of the clustering, which is not relevant in the result section since the applied clustering software is not their own development.

6. In most cases it is easy to follow the methods and understand the rationale for the decisions. However, in a few cases, some additional details should be provided, e.g. the weighting metric used for the MCL, the exact parametrization of MMSeqs cluster clustering and the exact calculation of the hypergeometric test in the case of GO (which values were compared to each other?).

7. Supplementary Fig 14 shows the stringent version of COG enrichment, however, a similar plot is not included about GO enrichment with the stringent criteria (50% threshold). The effect of this threshold could be more pronounced in the case of GO, especially among low c-values.

8. The gain and loss ratios (Figure 2) seem to follow a zig-zag pattern. Optionally, I think it would be worth checking whether it was influenced by the tree topology or not. For example, if loss predominates, the 'sister clade' contains only a small number of the species sampled.

minor points:

- "Shifts in the protein sequence space" in L 121-122, a more detailed explanation may help to understand what they mean here.
- According to the methods, or table legend it is not clear how "Gain tested" (in Supplementary Table 2) values are calculated. Could you specify it?
- In my opinion the argumentation for model species "adequate number of phylogenetic levels (phylostrata)" is unnecessary and at least partially independent thing in L841 and L252, but completeness of their functional annotation is much important argument, that the authors also mentioned.
- The connected components algorithm tends to minimize singleton clusters, but if they exist, how were they accounted for in the analysis? As a species-specific gene family in the last phylostratum for each species? I could not find any information on this.
- In Supplementary Figure S1. Supplement L9-10: labels and legend of pictures should be replaced: a and b are the plants and c, d are the fungi
- L269-275 seems to be introductory or methods but redundant in the result.
- L675-676 I am not sure that many losses in a relatively short time period indicate adaptive evolution. If a functionalities became unnecessary, like flagellum in fungi (L728-730), or the mentioned pathway (L677-679), all of the components might get lost in a relative short time period (the resolution is around millions of years) just because their function is not needed anymore (or change their functions).
- Version of MMSeqs2 in L862 is missing
- In Figure 1 the Color of the line C=4 is invisible.

RESPONSE TO REVIEWERS' COMMENTS

Reviewer #1 (Remarks to the Author):

I had already read a previous submission of the paper titled "Macroevolutionary dynamics of gene family gain and loss along multicellular eukaryotic lineages" by Domazet-Lošo and collaborators. I raised a series of concerns in the previous submission, that are not addressed in this one. I will share here my previous comments (some line numbers might be off).

We would like to thank the reviewer for the valuable comments on this and the previous submission of our paper. We very carefully considered and addressed all of them in detail here.

This study analyses the 667 complete genomes across the Tree of Life to profile the gene complements of different ancestors, going back to the last common ancestor of cellular life. The analyses show ancestral peaks of genome complexity followed by simplification, used to discuss the complexity by subtraction and the biphasic models of genome evolution, as well as showing that new genes display functional enrichment while functions gene losses seem to be evenly spread.

However, there is a major methodological concern that casts major doubts on the paper's findings. The comparative genomic analyses are anchored in four focal species, instead of performing evolutionary analyses in which all the species analysed are considered together. This approach is not up to the standards of the field, and it is reminiscent of phylostratigraphic approaches that have been often criticised in the literature. Focus on the genes of one species means that all the patterns of gains and losses specific to this lineage, from the origin of life to nowadays, are going to severely impact the reconstruction of the gene complements in ancestral nodes. For example, using the genes present in *Homo sapiens* to reconstruct the gene complement of the first bilaterian means that all the genes lost in the nodes for deuterostomes, chordata, olfactores, etc. will be omitted from the reconstruction of the bilaterian ancestral genome.

We fully agree with the reviewer that a comparative genomic analysis should consider all the species together and that this is a standard in the field. In our study we did exactly that, thus the criticism in this direction is misplaced. The total number of gene families (gene complements) for the ancestral nodes are reconstructed at once using the full phylogenetic tree and all reference genomes. This means that all available genomic information is considered in assessing the content of the ancestral genomes at all nodes.

To prevent any further ambiguity in this regard, we now improved presentation, fully resolved the phylogenetic tree, recalculated gain, loss and total gene family content at every node on the phylogeny where this calculation was possible (Supplementary Data 1, Supplementary Data 2). We also depicted changes in the gene family content from the perspective of every eukaryotic species, i.e., we treated each of the 352 eukaryotic taxa as a focal species (Supplementary Data 3). These additional analyses clearly showed that the increase-peak-decrease pattern is not a specificity of our four representative species, but a general pattern that holds essentially in all eukaryotic branches.

However, for the sake of presentation simplicity, in downstream functional analysis we retained focus on our four representative lineages. Similar approach was used in a recent high-profile paper (Ocaña-Pallarès et al. Nature 2022), where functional analyses were presented through trajectories of the two focal species (*H. sapiens* and *N. crassa*).

Proof of this is found in the same paper. For example, one of the figures shows the presence of cilium-related genes in the first eukaryote (LECA), and the number of these genes in LECA changes depending on the focal organisms used. This makes no biological sense. How many other genes have been missed due to only looking at these four species? We can reconstruct the ancestral genome of the first eukaryote using all the genomes of eukaryotes at once, there is no reason to limit analyses to four terminals. Evolutionary studies have to consider all the taxa studied and compare them all vs all, otherwise, they run into biases and artefacts. In a time in which most papers map gene gains and losses in the nodes of a phylogeny, considering hundreds of species at once, inferred using sophisticated methods, it is not adequate to draw major conclusions about major evolutionary transitions from only four species. What is the scientific rationale for this artificial constriction?

We respectfully note that this remark is based on the incorrect interpretation of Fig. 5 content. This misinterpretation probably hints that our description of functional enrichment analysis needs some improvements. As we already explained in the answer to the previous comment, we generated a gain-and-loss pattern in all-against-all manner for the all 667 reference species. We also fully agree with the reviewer that the number of genes (i.e., gene families) at LECA, or any other node on the phylogeny, should not change depending on the focal organism used. Accordingly, in our study these numbers are always the same (Supplementary Data 1,2). Moreover, the number of functional annotations assigned to particular clusters is also invariable and cannot depend on the choice of the focal species.

However, Fig. 5. does not show the number of cilium related gene families at LECA. Instead, it shows the statistical enrichments of cilium-related functional annotations at LECA. The aim of this analysis is to detect lineage-specific functional adaptations. The only way to achieve this is to look at a functional annotation along a particular lineage, and then to discern phylogenetic nodes that have unusually high concentration of that function. In this type of analysis, it does not make much biological sense to take other independent lineages as a reference, because that would collate the functional signals of many parallel evolutionary events, and consequently obscure lineage specific adaptations.

In this context, it is expected behavior that at a shared node an enrichment signal varies depending on the focal lineage. Although there is always the same number of cilium-related functional annotation at LECA, whether they are enriched or not will depend on the overall distribution of this functional annotation along all nodes that define a specific focal lineage. This is controlled by hypergeometric test.

For instance, in the yeast analysis enrichment signal at LECA (ps6, Fig. 5) is detected because in the later fungi-specific nodes there is no massive innovations related to cilium functions that would overshadow the LECA (ps6) signal. Actually, we correctly detected that fungi lost this functionality (Supplementary Data 12). In contrast, animal lineage experienced a massive gain of cilium-related functions at choanozoans (ps12, Fig. 5) which is a statistically more surprising event than their previous gain at LECA (ps6, Fig. 5).

To avoid any further confusion, we now added this explanation to the methods (L997-1016) and results (L512-L527) sections.

Another major issue is the conflation of gene complements in different ancestors with gene gains and losses. For example, Figure 1 shows ancestral gene complements which are used to justify the trajectory of genomic gains/losses across the Tree of Life. But ancestral complements do not reflect gains and losses. For example, a given node could gain 1000 gene families and lose another 1000, and the number of genes in Figure 1 wouldn't change. This argument is used to support different hypotheses around the role of gains and losses in different parts of the Tree of Life.

We agree with the reviewer that the number of gains and losses should not be conflated with the total number of gene families at the particular node (gene complements). Exactly for that reason we showed Fig. 2 and Supplementary Data 4. We now insured that this is clear throughout the text. We recovered all gain and loss patterns and then used these to calculate the total number of gene families (gene complements). All three values are now clearly presented in Supplementary Data 1 and 2.

However, we also note that gain and loss patterns in a particular node are rarely in balance, i.e., often one of the processes dominate (Fig. 2). This, in turn, implies that in the most instances increase (decrease) in the total number of gene families entails a dominant gene family gain (loss) in the background (L358-363).

Finally, at the methodological level, the authors criticise some of the current state-of-the-art methods (e.g., OrthoFinder) for not considering internal sequence conservation. Their approach using different c-values is interesting. However, they do not mention in their approaches any correction in the gene searches for gene length. The impact of gene length in BLAST searches is well-known, and important when looking at deep homology (e.g., older longer genes will be assigned to the correct gene family more often than shorter genes).

The c-value approach as implemented in MMSeqs2 inherently controls for possible gene length bias by forcing gene clustering to length bins. However, to additionally test could some resilient

gene-length bias influence our results, we now added gene length and phylogenetic distance normalization procedure to our MCL pipeline as described in the OrthoFinder paper (Emms and Kelly 2015). This analysis showed that OrthoFinder-type gene length normalization generally does not influence the evolutionary trajectory of protein family content (Supplementary Fig. 3 and 4). Please note that similar question was raised by Reviewer #2 (L282-292 Results, L951-958 Methods)

Similarly, the specific topology used can have a dramatic impact on Dollo parsimony reconstruction. The one provide in Supp File 1 shows many collapsed nodes, which cannot possibly have been used to infer gains and losses. For example, how is it possible to infer which genes are new or lost to animals using Dollo parsimony if the phylogeny does not resolve which is the first-splitting lineage of animals? What about LUCA?

The collapsed nodes certainly lower the resolution of gain and loss reconstruction in a respective lineage. However, they do not introduce inaccuracies. Actually, to accurately resolve which gene families are new or lost at the animal root no information on the later phylogenetic splitting within animals is needed. Initially, we took this advantage to avoid taking sides in phylogenetic disputes that have not yet been settled, e.g., ctenophore vs. porifera debate on the earliest diverging animal group.

Nevertheless, we now fully accepted the reviewer's suggestion to resolve all eukaryotic nodes on the phylogeny (including early diverging animals) by following the most recent phylogenetic literature (Supplementary Data 1, Supplementary References). Although these phylogeny improvements increased resolution, they did not change any of our initial findings (Fig. 1, Supplementary Data 3, etc.). This demonstrates that collapsed nodes on the side branches do not impact gain and loss pattern reconstruction along the focal lineages, as they act as outgroups. Please also note that we retained bacterial and some of the archaeal side branches unresolved as they serve only as outgroups on the path to eukaryotes.

Some other comments that I hope are of help to the authors:

1. The introduction picks on a few recent studies on animal evolution but neglects to mention many other ones around plants, fungi, and LECA published in the last few years. These studies have been published in high-impact journals. This is more shocking in the results and discussion sections, where large conclusions are drawn around the genomic changes in these nodes with no reference to other studies that showed the same patterns before.

We apologize for not including all relevant literature in the early version of this manuscript to which the reviewer had access to. However, in the Nat. Comm. version we carefully cited all papers that we found relevant for the story. To ensure that we covered all relevant papers in this revision we again searched the databases for important studies in high impact journals. We hope that we were successful in this regard, but we would also appreciate a direct suggestion if something is still missing.

2. Likewise, the article raises an interesting point about the different levels of clustering used in different studies. However, it fails to recognise the levels used in the various articles and misassigns them. For example, line 80 claims that some of these studies use orthogroups when some of these articles actually do use homology groups and they are even named as such in the papers.

We already addressed this remark in the version submitted to Nat. Comm. (L83-92), where we explained the current naming confusion related to clustering levels (i.e., orthogroups vs. homology groups). The reviewer correctly notes that Paps and Holland 2018 (and their follow-up papers) use the term 'homology groups' for their clusters. However, these papers use essentially the same pipeline (reciprocal-best hits plus MCL) as other similar studies in the field. However, in contrast to Paps and Holland 2018, the other studies designate their obtained clusters as 'orthogroups'. To double-check that this is indeed the case we contacted Jordi Paps (Uni. of Bristol) by email. He confirmed that their pipelines always filtered for reciprocal blast hits, basically meaning that they trace orthologs, although they named them as 'homology groups'.

3. The study does not dwell on the last common ancestor of all cellular life. Figure 1 shows 8k-12k

gene families in this ancestor, which means this ancestor of all life had at least that many genes if not more. This is possibly the most striking finding in the study but it is totally overlooked. Which are these gene families? How does this elevated number of gene families compare to other studies (e.g., recent papers by Bill Martin)?

Our study estimated the number of gene families that are shared between bacteria and archaea (Fig. 1, ps1). This value should not be equated with the content of gene families in LUCA, a number which is much harder to estimate due to the non-vertical inheritance during early diversification of cellular life (e.g., Weiss et al. 2016). That said, our estimate that bacteria and archaea share between 8k-11k gene families (Supplementary Data 2) matches quite well the estimate of the most recent Bill Martin's paper (Weiss et al. 2016) where they found that 11k protein families contained homologues from bacteria and archaea. We now adjusted the text accordingly (L303-311).

4. L253, how is it possible to get a gain/loss ratio for LUCA in Figures 2 and 3 if there are no outgroups for LUCA? In which other organisms are the genes lost in LUCA found? Shall not be the gain/loss ratio infinite?

We suspect that an oversight elicited this comment. We respectfully point to the fact that in Fig.2 and 3 we never showed gain/loss ratio for LUCA, neither we made any calculation of gene loss for LUCA (Supplementary Data 2). As the reviewer correctly note, this is not possible. Actually, we already explicitly explained this fact in the Methods section (L970-975). However, to help the readers immediately notice this property, we now additionally explained it in the captions of the relevant figures (Fig. 2,3).

5. Different parts of the paper (e.g., L86, 572, 577, etc) claim that the approach used here recovers more “macroevolutionary changes” or more “biologically significant” findings than other approaches. Which criteria are used to make these claims and to identify how significant are those changes compared to other studies?

We thank the reviewer for this comment. We already addressed it in the version submitted to Nat. Comm. We explained better that we refer to functional macroevolutionary signals. They could be recovered by statistical approaches (e.g., enrichment analysis), which uncover macroevolutionary biases by statistically controlling for noise. For instance, we quantified in this version how many GO functional terms show significant enrichments depending on the c-values choice (Fig. 10). Similar quantitative approaches could be used to compare different clustering pipelines including sequence similarity search algorithms, normalization steps and clustering algorithms. (L774-786, L841-849)

6. I disagree with the concept that all extant genes are homologous at some level, as all coalesce to the first gene that ever existed. That assumes that genes only emerge by gene duplication. This is surprising based on the pioneering work of the authors about *de novo* gene generation. Does it mean that *de novo* genes don't leave any descendants? It also ignores that new genes can emerge by gene fusion or fission.

This remark is outdated. In the submitted version we improved this part of the text. We, of course, agree with the reviewer that several mechanisms contribute to the formation of new genes (e.g., duplications, divergence, *de novo* emergence, reshuffling and combinations of them). We also fully agree with the reviewer that the diversity of these mechanisms makes a simplified models of gene evolution unrealistic. For exactly that reason we question the validity of the current simulations studies that use only speciation-divergence model which inherently ignores duplications-divergence and *de novo* gene evolution. Please see the current discussion on this subtopic (L828-840).

7. Similarly, L500 links the simplification of bacterial genomes in pangenome contexts with genome simplification in multicellular eukaryotes due to diet changes. This relation is far-fetched and would require evidence to support it.

We thank the reviewer for pointing this out. A very illustrative example in this regard is the loss of capability to synthesize nine amino acids (essential amino acids) at the root of animal tree. This massive metabolic simplification, which is a synapomorphy of animals, is most likely linked to the

diet changes that allowed the animal ancestor to efficiently get these essential amino acids from the environment. More efficient filter feeding and/or predation are innovations that probably led to this reduction. (L685-690, L724-728)

8. Related to the previous point, L525 suggests that organismal complexity could be estimated from the number of ancestral conserved functions plus the functions achieved through interactions with other organisms. How this applies to organisms such as parasites? A parasitic worm with no digestive tract or significant nervous system would be inferred as complex because most of the lost functions are performed via the host.

We thank the reviewer for suggesting this. Our concept of functional outsourcing could be applied to parasite-host interactions as well. An excellent example are microsporidians — fungi-related eukaryotic intracellular parasites of animals. The genomes of microsporidians underwent an extreme reduction due to obligate intracellular parasitic life style. However, not all lost molecular functions of microsporidians contribute equally to their genome complexity, if viewed from the lens of functional outsourcing.

Microsporidians lost mitochondrial respiration, including mitochondrial genome, however they evolved adaptations for harvesting ATPs produced by the mitochondria of host cells. This is clear example of functional outsourcing where an organism reduces a part of its genome but retains its benefits through biological (parasitic) interaction. This interaction secures that microsporidian ancestor which possessed mitochondria and the extant microsporidians that lack them have comparable complexity related to this particular function.

An opposite example is the reduction in the number of splicing-related genes in some extant microsporidians. In these microsporidians the number and diversity of splicing-related genes is severely reduced, because they compacted their genome and simplified their introns. As a result, a substantial part of the splicing machinery became obsolete and they lost it over time. However, this functionality was not outsourced because microsporidians do not rely on the splicing machinery of host cells. In sum, this means that the overall complexity of microsporidians related to splicing decreased. (L757-773)

9. L583 and the following criticise studies using simulations (Moyers, Zhang, Paschalis) for their lack of functional meaning. While this point is correct, these papers never claimed to provide functional findings. Their point is to highlight methodological issues, and they are really great at doing so.

We respectfully note that this remark does not reflect the literature. A central analysis in Moyers and Zhang 2015 paper tests whether simulated sequences could produce statistically significant functional signals, i.e., germ layers related macroevolutionary signals in *Drosophila*. By default, such functional signals are not expected in their simulations unless some technical bias generates it. Surprisingly, the authors initially reported the existence of such functional signals, suggesting that they were produced by the limitations of the sequence similarity search tools.

However, this analysis turned out to be flawed due to a calculation/programming error. After realizing this error, the same authors published a citable corrigendum where they retracted all their results and claims related to the functional signals (Moyers and Zhang 2016, <https://doi.org/10.1093/molbev/msw202>). This is an obvious example that functional analysis was attempted in at least some of these studies. We now explained this in the discussion more precisely (L823-827).

10. Methods claim to use BUSCO to filter genomes based on quality, however, there is no mention of the completeness threshold used to select such genomes.

In this version, we updated the BUSCO analysis to ensure the overall quality of genomes (contaminations, completeness, duplications - Supplementary Data 16). We thus provide in Methods a detailed description of the new BUSCO protocol. A single completeness threshold would be rather crude approach for such a heterogeneous and large set of reference genomes that would tend to unnecessarily eliminate parasitic and highly derived organisms. We thus applied a stepwise evaluation protocol that considers all BUSCO parameters and species biology (Manni et al. 2021 Curr. Protoc.). Please see also answer to the similar comment of the Reviewer #3. (L889-911)

Reviewer #2 (Remarks to the Author):

Review

In the manuscript “Macroevolutionary dynamics of gene family gain and loss along multicellular eukaryotic” by Domazet-Lošo et al., the authors infer putative orthology across a large sampling of Eukaryotic genomes, aiming to reconstruct the history of gene gain/loss patterns. This analysis focuses on the use of four focal species, which are well-known model organisms from animals, plants, and fungi. Although the authors examine the phylostratigraphy of additional taxa (e.g., the truffle), the use of such few ‘focal’ species to represent the evolutionary dynamics of entire kingdoms seems a bit misleading. I think that are great reasons to use these ‘focal’ species as controls, but insights into other lineages with interestingly evolutionary dynamics would bolster the impact of the manuscript.

We thank the reviewer for very constructive comments. We agree that it is much better to show results from the perspective of all eukaryotic taxa. In the revised version, we present gene family gains, losses and totals for the eukaryotic part of the tree (Supplementary Data 1,2). In addition, we show ancestral gene family trajectories for the 352 eukaryotic taxa, i.e., we treat every eukaryote as a focal species (Supplementary Data 3).

Major comments

1) Line 20: To say four focal species represent the diversity of three kingdoms is a stretch. Could this be rephrased throughout?

We rephrased this to reflect the fact that we now provide gain, loss and gene family total analyses for all eukaryotic taxa.

2) As a control, does your method identify known whole genome duplication events, such as the ones in budding yeast or flowering plants?

As we trace strictly homologs, our method is not generally sensitive to whole genome duplications. Duplicated genes, no matter how numerous they are, will be simply clustered into parental gene families as long as they retain sequence similarity. However, we can eventually detect unusually high rates of novel gene generation, which are probably underpinned by genome duplication or hybridization events. An example of such events is the case of Brassica species (Supplementary Fig. 1, Supplementary Data 3). The Brassiceae tribe seems to be a phylogenetic hot-spot for generation of new gene families probably related to allopolyploid hybridization events. We added this example in the Results (L189-199).

3) Much of the gene family analysis will depend on taxon sampling. Can the authors provide a figure to detail the depth of taxon sampling at each ps-level?

We now provided a fully resolved tree, where all reference taxa are visible for every node on the phylogeny (ps-level) (Supplementary Data 1 and 2).

4) More broadly, the meaning of the ps-levels isn't very clear. For example, are the steep increase and decreases due to a confounding of large geologic time between taxonomic ranks? Inversely, couldn't the lack of pronounced peaks at ps 23-35 in the phylostratum of *S. cerevisiae* be largely attributed to a lack of evolutionary time between each taxonomic rank? Although taxonomic ranks are informative, and a reasonable x-axis for figure 1, I think the time-component is overlooked and should somehow be accounted for, or at least discussed.

We agree that ps-ranks (nodes) show topological levels on the phylogeny, which means that only ordinal time, and not absolute time, can be assessed in this representation type. This entails that we can discuss differences in the number of protein families between ps-ranks, but cannot relate to their rates of change; i.e., we cannot assess how fast that change happened.

The evolutionary time between our phylostrata have tendency to be shorter towards more recent phylostrata, however this does not influence the recovered profile trends. The reason for this

comes from the fact that switching to the absolute time on the x-axis would not change any value on the y-axis, it would only stretch or shrink the distance between some points along x-axis.

The origin of Saccharomycetaceae (ps24, Fig. 1c) was estimated to about 150 Mya. This absolute time roughly corresponds in the plant lineage to the origin of Eudicots (ps19, Fig. 1d). From that time point to the present, the plant lineage experienced many detectable changes in the number of protein families (ps19-ps32, Fig. 1d). This demonstrates that the yeast lineage also had in principle enough evolutionary time to experience more dramatic changes in the number of protein families. Actually, the example of Brassica lineage discussed in the previous answer shows that we can detect changes in the number of gene families for internodes which are rather short, i.e., less than 10 My.

We now added a paragraph along these lines in the Results section (L200-215).

5) The authors did a nice job comparing MMseqs2 to MCL-based clustering. Advances in orthology inference — such as implemented in OrthoFinder — also account for length-related biases in sequence similarity scores. Not accounting for these biases results in oversplitting orthologous groups of genes. Does MMseqs2 also conduct this correction? If not, this may explain the larger number of gene families observed in, for example, Figure S3 and the root of Eukaryota.

This is an excellent point, also raised by Reviewer #1. The c-value approach as implemented in MMSeqs2 inherently controls for possible gene length bias by forcing gene clustering to length bins. However, to additionally test could some resilient gene-length bias influence our results, we now added gene length and phylogenetic distance normalization procedure to our MCL pipeline as described in the OrthoFinder paper (Emms and Kelly 2015). This analysis showed that the OrthoFinder-type gene length normalization generally does not influence the evolutionary trajectory of protein family content (Supplementary Fig. 3). In principle, not accounting for gene length biases could result in an over-splitting of homologous groups. However, our test showed that this effect is negligible, especially for high c-values (Supplementary Fig. 4). Moreover, the MMseqs2 clustering turned out to return less clusters compared to the MCL clustering either with or without gene length normalization (Supplementary Fig. 4). We now discuss this test in the Results section (L282-292).

6) The analysis of lineage-specific genes may be complicated by the use of different gene annotations predicted using different software. See Weisman et al. <https://www.sciencedirect.com/science/article/pii/S0960982222007217>. This may be worth noting in the discussion.

Thank you for pointing out this paper. We now discuss that annotation heterogeneity could complicate the analysis of lineage specific genes, especially in relatively closely related species, whereas in distantly related species and in large sets of genomes this effect should be less pronounced (L850-856).

7) Can the authors provide greater detail about the Dollo parsimony-based algorithm employed for their analysis? It is not clear what was used in the GitHub repo to do that analysis.

We now improved documentation and commenting in our custom-made software to clearly indicate where the Dollo's parsimony-based algorithm was implemented (GitHub files getGL.py and nodesGTL.py). We also better described applied procedure in the Methods section (L959-970).

Minor comments

1) Can the authors clarify what they mean by “if the quality looked compromised” on line 857?

In this version we updated and rerun the BUSCO analysis to ensure the overall quality of genomes (contaminations, completeness, duplications). We thus provide a detailed description in the Methods of the new BUSCO protocol. Please see also answer to the similar comment of the Reviewer #3. (L889-911)

2) Version information is provided for some, not all software. For reproducibility, can this information be included in the manuscript?

We included version information for all software in this latest version of the manuscript.

Reviewer #3 (Remarks to the Author):

Domazet-Lošo et al. have profiled the genomic content and macroevolutionary trajectories from the ancestor of cellular organisms to four model organisms (two animals, one plant and one fungus). To achieve sufficiently detailed clustering and high resolution for phylostratigraphy, they included 676 whole genomes in the analysis. The results are robust, which could be partly due to this impressive dataset. The other reason could be the clustering approach, which explores a high range of possible scenarios without choosing an arbitrary threshold, even though the chosen clustering approach may be burdened with biases. Another indisputable advantage of the study is that they seek to extend the analysis to deeper nodes in the tree of life than ever before. They found that gene family content follows a common pattern between these different model species. The number of gene families reaches a high value (genomic complexity) from which we can observe a gradual decrease towards the extant species. Based on the current literature, this is the expected pattern, but the generality and robustness of the phenomenon presented here would be a significant contribution to the field. Furthermore, they provide exciting hypotheses for the existence of reductive evolution (ecological specialization and functional outsourcing), although these hypotheses are based mainly on the literature and not proven by their own results. The paper is carefully drafted, the approach and methodology is adequate but might necessitate some care/reconsideration. Based on the highlighted examples the COG and GO enrichment also seems to be reliable, but methodological questions are raised with this from my side.

Overall, I think this article would represent a considerable step in understanding the gene family evolution of eukaryotes. However, I see some potential methodological weaknesses that should be addressed during the revision process, in order to rule out the possibility of biases.

We thank the reviewer for the supportive assessment and constructive comments that helped us to improve our manuscript.

major points:

1. One of the most obvious advantages of the study is using a large number of genomes to determine the most accurate cluster membership and phylostratigraphic resolution. Nevertheless, the authors failed to exercise reasonable caution about the quality of the genomes used, which can greatly affect the analyses of genome evolution, especially the ancestral genome reconstruction. The contamination of genomes could lead to an overestimation of gene content in ancestors and inflates the rate of gene loss, as shown by Balint et al. (2022: <https://www.biorxiv.org/content/10.1101/2022.11.17.516887v1.full.pdf+html>). For example, the genome of *Quercus suber* contains an entire fungal genome as contamination, resulting in a >6000 (cca. 60%) copy number overestimation for the LECA in the study Balint et al. (2022). *Quercus suber* is also listed in the Supplementary Table 1 of Domazet-Lošo et al. Biased overestimation of LECA is consistent with the phenomena presented by the authors in Fig 1 (higher gene family content in ancient nodes and gradual loss towards the recent nodes). Therefore, I strongly recommend the authors to recheck/clean up the genomes used to exclude this bias, or at least get a picture about the extent of the bias.

This is indeed an important remark. We fully agree that the eventual contamination of reference genomes could induce biases and pull the origin of protein families to the ancestral nodes. We now state this and cite the suggested reference. To confirm that our reference genomes do not contain obvious contaminations, we applied a multistep approach in this version.

To ensure that our dataset is free from cross-lineage contaminations we applied a BUSCO contamination detection protocol. In this analysis, we independently searched every genome using six different BUSCO lineage datasets (bacteria, archaea, eukaryota, metazoa, viridiplantae and fungi). We then looked for genomes that had unusually high BUSCO scores in lineages other than the native one. This inspection revealed that the two bacterial, sperm whale and *Quercus suber* genomes had comparably high BUSCO scores outside their own lineage. We excluded these genomes from the analysis and included non-contaminated genome versions, either coming from the same or closely related species.

In addition, to detect eventual within-lineage contaminations, we individually evaluated every genome that had BUSCO duplication score above 20%. In some cases, high duplication scores are expected due to known whole genome duplications events (e.g., Brassica species). However, in other cases we substituted these genomes with better versions that have lower duplications scores.

Finally, we manually checked every genome to see if in the source database a respective genome has been flagged as possibly contaminated. However, no further cases were detected. We described this procedure in the Methods section (L889-911).

After we performed all of these updates to our reference genome dataset, we re-run all analyses in the manuscript. The obtained results are highly similar to the original findings, which demonstrates that the recovered biological patterns are fairly robust to occasional contaminations of reference genomes.

2. Horizontal gene transfers and incorrect gene annotation (e.g. artificial fusion proteins especially at low c-values) can also influence the phylostratigraphy, although not to such an extent as genome contamination. HGT was mentioned once (L632-634), but these limitations of the approach should be emphasized more strongly, e.g. in the methods or in the discussion.

This is true. Similar to genome contamination, horizontal gene transfer (HGT) can also pull the origin of gene families to older nodes on the phylogeny. However, HGT in eukaryotes is generally less frequent than in prokaryotes, thus these effects should have lower impact on the eukaryotic part of the phylogeny. Similarly, the impact of HGT will also depend on the phylogenetic distribution of a gene family. For instance, the phylogenetic origin of a relatively ubiquitous gene family will be less sensitive to HGT than the phylogenetic origin of a gene family which is specific for a narrow lineage. In this version, we explained this more strongly in the Discussion section (L856-864).

3. The clustering approach touches on extreme setups (connected component c-value=0), therefore the following descriptive statistics on clusters are worth to be prepared as graphs or tables and attached as supplementary files. 1) Distribution of cluster sizes, similar to Ocaña-Pallarès et al. (Nature, 2022 Supplementary Information 2-Fig. 7.). This would be useful for assessing the reconstruction of the ancestral genome of LECA (L293-307). 2) Domain and/or GO diversity of clusters, which would be useful for understanding the relationship between functional signal and alignment length/clustering strategies (Figure 10). I have the impression that simply the higher number of clusters in strict clustering (c-value=0.8) could lead to an increase in the number of significantly enriched GO categories. 3) Pairwise distribution of cluster coverage for each clustering setup (different c-values). A pairwise coverage threshold is not necessarily true for all components of a cluster. Therefore, the statement "c-value = 0.8 which forces at least 80% of sequence length alignment overlap" can be applied to the pairwise protein pairs but not to the final clusters.

Thank you for suggesting additional descriptive statistics, this turned out to be quite helpful. 1) We now presented distribution of cluster sizes in the relation to the percentage of clustered sequences, akin Ocaña-Pallarès et al. Nature, 2022 (Supplementary Fig. 12). This analysis showed that low c-values are pushing sequences into mega-clusters mainly at the expense of moderately sized clusters. We see this pattern as an additional argument why we recover less functional signals at the lower c-values. We agree with the reviewer that the overall higher number of clusters in strict clustering (c-value=0.8) contributes to an increase in the number of significantly enriched GO categories. However, this factor must be coupled with cluster-size distribution change under permissive clustering (c-value=0).

2) We calculated the average number of GO categories per cluster in relation to cluster sizes (Supplementary Figure 13). It is evident that the permissive clustering (c-value = 0) decreases the percentage of GO annotations in small to middle size clusters. This is probably yet another reason why we recovered less significantly enriched GO categories in the permissive clustering (c-value = 0).

3) We agree that our wording was not fully precise here. According to the MMSeg2 documentation, c-value = 0.8 forces at least 80% sequence length alignment overlap between a 'representative sequence' of a cluster and all matching cluster members. Since a representative sequence (a sort of centroid) has the best interconnection within a cluster and was used as a seed to construct the cluster, we think that recalculating pairwise coverage for all sequence pairs within the cluster would not necessarily be more informative measure of alignment coverage. We adjusted the text along these lines (L807-817, L126, L160, L923).

5. Comparing clustering approaches (connected component vs MCL) is a useful and essential part of the work. However, as it stands, paragraph L293-307 is not so convincing, or partially unnecessary. In the first part of this paragraph, the authors argue that the gene family content of LECA was estimated to be 12,500 clusters (with the most permissive clustering, c -value=0), which number is close to the recently published estimation: 10,233 Pfam domains in LECA. Therefore, - they argued - the chosen clustering approach is better than MCL clustering which estimated 24-27,000 clusters. Since the authors have not shown/proven the domain homogeneity of the most permissive clustering (c -value=0) the two values (number of clusters and number of Pfam domains) should not be compared to each other. Also the authors state elsewhere that attraction of multidomain proteins in large clusters is an issue (L156-157). The second part of this paragraph (L302-L307) deals with the computational speed of the clustering, which is not relevant in the result section since the applied clustering software is not their own development.

We rephrased this part by providing more accurate LECA estimates that can be better compared to our results. We also removed the remark related to the computational speed of the clustering (L293-302).

6. In most cases it is easy to follow the methods and understand the rationale for the decisions. However, in a few cases, some additional details should be provided, e.g. the weighting metric used for the MCL, the exact parametrization of MMSeqs cluster clustering and the exact calculation of the hypergeometric test in the case of GO (which values were compared to each other?).

We now provided MCL weighting metrics, the exact parametrization of MMSeqs *cluster* and the exact calculation of the hypergeometric test in GO functional analysis (Methods, L912-915, L944-958, L997-1016).

7. Supplementary Fig 14 shows the stringent version of COG enrichment, however, a similar plot is not included about GO enrichment with the stringent criteria (50% threshold). The effect of this threshold could be more pronounced in the case of GO, especially among low c -values.

We now added the same analysis for GO enrichment, i.e., we applied the 50% threshold to GO functions. The results of this stringent analysis for *H. sapiens* are shown in Supplementary Data 17 (gain) and Supplementary Data 18 (loss). As the reviewer anticipated, lowering the number of GO annotations reduced the power of hypergeometric test. As a result, the total number of enriched GO functions dropped to 35%, especially affecting enrichments at lower c -values. However, essentially all the GO enrichments that we discuss in the paper are again recovered and their qualitative pattern stayed unchanged. We adjusted text along these lines (L981-992).

8. The gain and loss ratios (Figure 2) seem to follow a zig-zag pattern. Optionally, I think it would be worth checking whether it was influenced by the tree topology or not. For example, if loss predominates, the 'sister clade' contains only a small number of the species sampled.

We also noticed this zig-zag pattern. However, we think that is not related to the tree topology as it remained stable after we resolved all branching patterns in this version. For instance, the massive loss of genes at Metazoa could not be explained by the number of species sampled because both relevant sister clades (Metazoa and Choanoflagellida) are well populated. Rather we think that for some reason organisms alternate between the predominantly expanding and reductive mode of protein family evolution, i.e., rarely these two modes are in balance. We now note this in the results and suggest that possible mechanisms behind this pattern should be explored in future studies (L358-363).

minor points:

· "Shifts in the protein sequence space" in L 121-122, a more detailed explanation may help to understand what they mean here.

This stems from our previous work on orphan gene evolution and relates to the duplication-divergence model of novel gene evolution. We adjusted the sentence to reflect this.

- According to the methods, or table legend it is not clear how “Gain tested” (in Supplementary Table 2) values are calculated. Could you specify it?

The "gain tested" values relate to the total number of GO terms assigned to our clusters in a respective lineage. All of these terms were tested for enrichment in the respective focal lineages using hypergeometric test. We added explanation in this table which is now Supplementary Data 14.

- In my opinion the argumentation for model species “adequate number of phylogenetic levels (phylostrata)” is unnecessary and at least partially independent thing in L841 and L252, but completeness of their functional annotation is much important argument, that the authors also mentioned.

We excluded the argumentation related to the number of phylogenetic levels in the Results section, partially because we now also present every eukaryotic taxon as a focal species.

- The connected components algorithm tends to minimize singleton clusters, but if they exist, how were they accounted for in the analysis? As a species-specific gene family in the last phylostratum for each species? I could not find any information on this.

We retained only those species-specific gene families that contain at least two members, i.e., singleton clusters in the last phylostratum are ignored as one could argue that they are not yet a true gene family. We added a sentence that explains this to the Methods section (L963-965).

- In Supplementary Figure S1. Supplement L9-10: labels and legend of pictures should be replaced: a and b are the plants and c, d are the fungi

As we present in this version the total number of gene families for all eukaryotic species we omitted Supplementary Figure S1.

- L269-275 seems to be introductory or methods but redundant in the result.

We removed this part.

- L675-676 I am not sure that many losses in a relatively short time period indicate adaptive evolution. If a functionalities became unnecessary, like flagellum in fungi (L728-730), or the mentioned pathway (L677-679), all of the components might get lost in a relative short time period (the resolution is around millions of years) just because their function is not needed anymore (or change their functions).

We toned down this part.

- Version of MMSseqs2 in L862 is missing

We used the MMSseqs2 version 14-7e284.

- In Figure 1 the Color of the line C=4 is invisible.

We changed the color of the line to a darker shade.

REVIEWERS' COMMENTS

Reviewer #1 (Remarks to the Author):

I have re-reviewed the manuscript titled "Macroevolutionary dynamics of gene family gain and loss along multicellular eukaryotic lineages" by Domazet-Lošo and collaborators. The authors have demonstrated a willingness to address my concerns by clarifying their methods (and my errors!), correcting inaccuracies, providing additional data, and revising their interpretations and discussions. I want to congratulate them for their efforts.

In some areas, like the use of a few focal species even if the authors claim this is just for clarity, we will have to agree to disagree. Overall, I believe that the paper is now in good shape, and it will undoubtedly spark further discussion and debate among the scientific community. I am confident that the community will provide valuable insights that will help to advance our understanding of these complex and fascinating issues.

Reviewer #3 (Remarks to the Author):

I would like to thank the authors for incorporating the modifications into the manuscript. I appreciate the additional comparative analysis, which helps to better understand the clustering. I have no further objections to the manuscript.

RESPONSE TO REVIEWERS' COMMENTS

Reviewer #1 (Remarks to the Author):

I have re-reviewed the manuscript titled "Macroevolutionary dynamics of gene family gain and loss along multicellular eukaryotic lineages" by Domazet-Lošo and collaborators. The authors have demonstrated a willingness to address my concerns by clarifying their methods (and my errors!), correcting inaccuracies, providing additional data, and revising their interpretations and discussions. I want to congratulate them for their efforts.

In some areas, like the use of a few focal species even if the authors claim this is just for clarity, we will have to agree to disagree. Overall, I believe that the paper is now in good shape, and it will undoubtedly spark further discussion and debate among the scientific community. I am confident that the community will provide valuable insights that will help to advance our understanding of these complex and fascinating issues.

We thank the reviewer for the critical but constructive comments, which helped us to rethink and further improve our analyses and ideas. We share with the reviewer fascination on this topic and agree that some rare remaining points of disagreement are important building blocks that will help us all to advance the field.

Reviewer #3 (Remarks to the Author):

I would like to thank the authors for incorporating the modifications into the manuscript. I appreciate the additional comparative analysis, which helps to better understand the clustering. I have no further objections to the manuscript.

We thank the reviewer for very insightful suggestions and for recognizing our efforts to clarify the clustering procedure.